# FLATTENGPT: DEPTH COMPRESSION FOR TRANSFORMER WITH LAYER FLATTENING

## ABSTRACT

This work proposes **FlattenGPT**, a novel depth compression method for transformers. Recent works have observed redundancy across transformer blocks, prompting the research of depth compression to prune less crucial blocks. However, existing works mostly follow the entire-block pruning paradigm and suffer from risks of discarding knowledge learned in those blocks, leading to substantial performance degradation. On the other hand, channel pruning can better preserve performance, while it cannot compress model depth and is challenged by inconsistent pruning ratios for each layer. To address those issues, this paper introduces a novel compression strategy named layer flattening, which bridges the gap between layer pruning and channel pruning. By converting two adjacent blocks into one, it compresses the network depth and enables more effective parameter redundancy detection and removal. FlattenGPT strives to preserve the knowledge learned in all blocks and remain consistent with the original architecture, enhancing model efficiency with a decent trade-off to performance. Extensive experiments demonstrate that FlattenGPT outperforms existing pruning methods in both zero-shot accuracies and WikiText-2 perplexity across various model types and parameter sizes. It also outperforms other pruning methods in accelerating LLM inference, making it a promising approach for enhancing the efficiency of transformers.

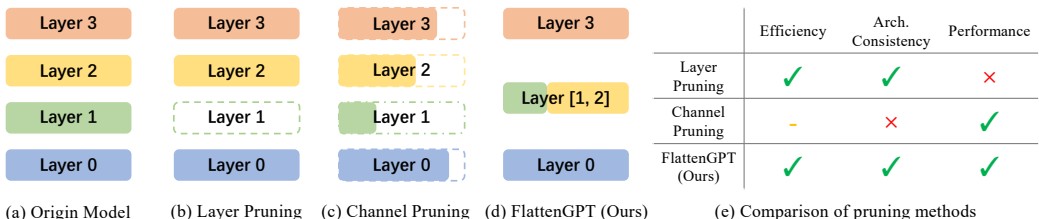

| | Efficiency | Arch. Consistency | Performance |
|---|---|---|---|
| Layer Pruning | ✓ | ✓ | ✗ |
| Channel Pruning | - | ✗ | ✓ |
| FlattenGPT (Ours) | ✓ | ✓ | ✓ |

(a) Origin Model    (b) Layer Pruning    (c) Channel Pruning    (d) FlattenGPT (Ours)    (e) Comparison of pruning methods

Figure 1: Comparison of pruning methods. (a) The original architecture. (b) Layer pruning removes the entire block and discards all knowledge in it. (c) Channel pruning cannot compress model depth and leads to inconsistent architecture across layers. (d) Our method bridges the gap, producing a compact model with little performance degradation. (e) A comprehensive comparison.

## 1 INTRODUCTION

Recent advancements in Large Language Models (LLMs) (Brown et al., 2020; Zhang et al., 2022; Chowdhery et al., 2023; Touvron et al., 2023a;b; Dubey et al., 2024) have led to breakthroughs in understanding and generation of natural language (Hadi et al., 2023; Zhao et al., 2023; Minaee et al., 2024). However, the cost of heavy computation and extremely large memory consumption makes it very challenging to deploy on resource-limited devices. To mitigate these issues, model compression has emerged as a popular post-training solution, reducing model size and complexity by removing model redundancy (Gupta & Agrawal, 2022; Zhu et al., 2023).

Depth compression (Song et al., 2024; Men et al., 2024) is a technique aimed at reducing the redundancy across transformer blocks. This redundancy manifests itself in the cross-layer similarity (Gromov et al., 2024; Sun et al., 2024; 2025). Figure 2(a) illustrates that the input of adjacent

blocks has high similarity in LLMs, which is caused by the residual path spanning the entire LLM. This similarity is particularly evident in LLMs, indicating that there is a certain amount of redundancy within them. Depth compression methods aim to reduce this cross-layer redundancy to achieve a compact network architecture. Besides, compared to other pruning methods such as channel pruning (Ma et al., 2023; Ashkboos et al., 2024) or 2:4 pruning (Frantar & Alistarh, 2023; Sun et al., 2023), depth compression methods have an evident advantage in inference speed with the same number of parameters (Song et al., 2024).However, previous depth comparison methods usually adopt layer or block pruning, which removes the entire block selected by measuring how crucial the blocks are (Men et al., 2024; Samragh et al., 2023; Kim et al., 2024; Song et al., 2024; Zhong et al., 2024; Zhang et al., 2024a). It may also remove the useful knowledge learned in the pruned blocks simultaneously, leading to serious performance degradation.

Channel pruning (Ma et al., 2023; Ashkboos et al., 2024; van der Ouderaa et al., 2024; Lin et al., 2024), on the other side, conducts a fine-grained parameter preservation and thus leads to better performance. However, these methods usually assign different pruning ratio for each layer. This inconsistency in module architecture will cause inconvenience in hyperparameter tuning or model deployment, such as LoRA hyperparameters (Hu et al., 2022). Moreover, channel pruning cannot utilize the redundancy across layers, resulting in a deeper architecture and higher latency in practice.

In this paper, we propose a fine-grained depth compression method called FlattenGPT, which preserves crucial knowledge while reducing the model depth. FlattenGPT is composed of two stages. In the first stage, we propose a new operation named flattening, to merge adjacent transformer blocks by concatenating their parameters and hidden states. This operation changes the sequential execution of transformer blocks to parallel execution, with only the input of the blocks being altered. Since the input features of each layer in LLMs are inherently of high similarity, flattening the blocks has little impact on the model's performance. The subsequent stage employs a channel pruning method to streamline the merged transformer blocks. Channel pruning can identify the critical channels within the merged blocks, allowing for a fine-grained removal of redundancy while preserving the learned knowledge of each block.

FlattenGPT has clear advantages over previous pruning methods. As shown in Figure 1, unlike layer pruning methods, flattening preserves the knowledge embedded in each layer, raising the performance ceiling of the depth compression. Compared with channel pruning, FlattenGPT produces a consistent architecture with lower depth, leading to higher efficiency and easier tuning and deployment. This method bridges the gap between depth compression and channel pruning, allowing for a more comprehensive model compression. Extensive experiments demonstrate that FlattenGPT preserves up to 96% of zero-shot performance with a compression rate of 20% on LLaMA 2 (Touvron et al., 2023b), outperforming prior depth compression approaches. To the best of our knowledge, FlattenGPT is an original effort on transformer compression through layer flattening. It shows potential to establish a novel comprehensive framework that enhances the depth compression of transformer architectures.

## 2 PRELIMINARY AND ANALYSIS

### 2.1 PRELIMINARY OF TRANSFORMER ARCHITECTURE

The Pre-LN transformer architecture in LLMs (Touvron et al., 2023a) consists of multiple decoder layers, each composed of two blocks, *i.e.*, Multi-Head Attention (MHA) and Multi Layer Perceptron (MLP). Concretely, let $l \in \{0, 1, \cdots, L-1\}$ denote the layer index, $T$, $d_h$, $d_{int}$ and $H$ denote the sequence length, hidden dimension, intermediate dimension, and the number of attention heads, respectively. The formulation of a Transformer layer is denoted as

$$\tilde{\boldsymbol{H}}^{\ell-1} = \boldsymbol{H}^{\ell-1} + \text{MHA}^{\ell}\left(\text{LN}_a^{\ell}\left(\boldsymbol{H}^{\ell-1}\right)\right), \boldsymbol{H}^{\ell} = \tilde{\boldsymbol{H}}^{\ell-1} + \text{MLP}^{\ell}\left(\text{LN}_p^{\ell}\left(\tilde{\boldsymbol{H}}^{\ell-1}\right)\right), \quad (1)$$

where $\boldsymbol{H}^{\ell} \in \mathbb{R}^{T \times d_h}$ denotes the output of the $l$-th layer, $\text{MHA}^{\ell}$, $\text{MLP}^{\ell}$, $\text{LN}_a^{\ell}$, and $\text{LN}_p^{\ell}$ denote the MHA block, MLP block, MHA normalization, and MLP normalization of the $l$-th Transformer layer, respectively. The normalization layers are usually composed of a root mean square normalization and an element-wise affinement:

$$\text{LN}_a^{\ell}\left(\boldsymbol{X}\right) = \text{RMSNorm}\left(\boldsymbol{X}\right)\text{diag}\left(\boldsymbol{\alpha}_a^{\ell}\right), \text{LN}_p^{\ell}\left(\boldsymbol{X}\right) = \text{RMSNorm}\left(\boldsymbol{X}\right)\text{diag}\left(\boldsymbol{\alpha}_p^{\ell}\right), \quad (2)$$

where $\text{RMSNorm}\left(\boldsymbol{X}\right)$ applies $\boldsymbol{X} \leftarrow \boldsymbol{X}/\|\boldsymbol{X}\|$ to each row of $\boldsymbol{X}$, $\boldsymbol{\alpha}_a \in \mathbb{R}^{d_h}$ and $\boldsymbol{\alpha}_p \in \mathbb{R}^{d_h}$ are the parameters of affinement.

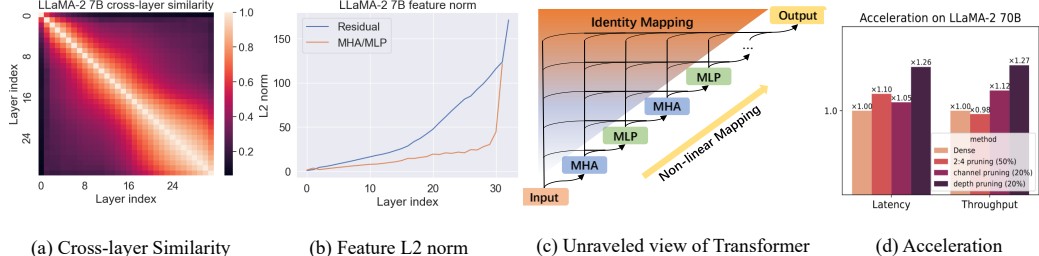

(a) Cross-layer Similarity    (b) Feature L2 norm    (c) Unraveled view of Transformer    (d) Acceleration

Figure 2: Redundancy in transformer blocks. (a) LLaMA-2 7B exhibits high cross-layer similarity. (b) The scale of the residual path grows faster than the MHA/MLP blocks, which dominates the deep hidden states. (c) The unraveled view of transformer architecture, where the residual path traversing the entire network leads to the cross-layer similarity. (d) The acceleration comparison between different pruning methods.

**MHA block**  The MHA block is defined as

$$
\mathrm{MHA}^\ell\left(\boldsymbol{X}\right) = \sum_{i=1}^{H} \mathrm{Softmax}\left(\sigma_r\left(\boldsymbol{X}\boldsymbol{W}_{Q,i}^\ell\right)\sigma_r^\top\left(\boldsymbol{X}\boldsymbol{W}_{K,i}^\ell\right)\right)\boldsymbol{X}\boldsymbol{W}_{V,i}^\ell\boldsymbol{W}_{O,i}^\ell, \tag{3}
$$

where $\boldsymbol{X} \in \mathbb{R}^{T \times d_h}$ denotes the input feature, $\boldsymbol{W}_{Q,i}^\ell, \boldsymbol{W}_{K,i}^\ell, \boldsymbol{W}_{V,i}^\ell \in \mathbb{R}^{d_h \times \frac{d_h}{H}}$, and $\boldsymbol{W}_{O,i}^\ell \in \mathbb{R}^{\frac{d_h}{H} \times d_h}$ denote the query, key, value, and output matrices of the $i$-th head in the $l$-th layer, respectively. For similicity, we denote $\boldsymbol{W}_Q = [\boldsymbol{W}_{Q,1}\ \boldsymbol{W}_{Q,2}\ \cdots\ \boldsymbol{W}_{Q,H}]$ as the horizontal concatenation of query parameters from all heads, and similar to $\boldsymbol{W}_K$ and $\boldsymbol{W}_V$. We denote $\boldsymbol{W}_O = \left[\boldsymbol{W}_{O,1}^\top\ \boldsymbol{W}_{O,2}^\top\ \cdots\ \boldsymbol{W}_{O,H}^\top\right]^\top$ as the vertical concatenation of output parameters. $\sigma_r$ denotes the positional embedding function.

**MLP block**  The MLP block is defined as

$$
\mathrm{MLP}^\ell\left(\boldsymbol{X}\right) = \sigma_s\left(\boldsymbol{X}\boldsymbol{W}_U^\ell\right)\boldsymbol{W}_D^\ell, \tag{4}
$$

where $\boldsymbol{W}_U^\ell \in \mathbb{R}^{d_h \times d_{int}}$ and $\boldsymbol{W}_D^\ell \in \mathbb{R}^{d_{int} \times d_h}$ denotes the up and down matrix and $\sigma_s$ is the non-linear activation function. $\boldsymbol{X} \in \mathbb{R}^{T \times d}$ is the input matrix. Prevailing LLMs (Touvron et al., 2023a;b; Bai et al., 2023) employ a gated MLP. Its up matrix is composed of a up matrix and gate matrix $\boldsymbol{W}_U^\ell = [\boldsymbol{W}_u\ \boldsymbol{W}_g]$, where the non-linear function is defined as $\sigma_s\left(\boldsymbol{X}\boldsymbol{W}_U^\ell\right) = \boldsymbol{X}\boldsymbol{W}_u^\ell \odot \sigma_g\left(\boldsymbol{X}\boldsymbol{W}_g^\ell\right)$. For the following discussions, we take the gated MLP as the baseline architecture.

## 2.2 ANALYSIS ON THE REDUNDANCY IN DEPTH

As illustrated in Figure 2(a), deep transformer architecture exhibits high cross-layer similarity. This is caused by the curse of depth (Sun et al., 2025), which implies that the deep layers are dominated by the residual path, *i.e.*, identity mapping. As shown in Figure 2(b), the L2 norm of the residual path is much larger than the MHA/MLP output in deep layers, dominating the forward propagation. An intuitive interpretation is shown in the triangle-shaped unraveled view of transformer architecture in Figure 2(c). The amount of residual features increases in deep layers and surpasses the non-linear blocks, leading to approximately identity mapping. This analysis shows the cross-layer redundancy in the transformers.

We provide a theoretical analysis of layer redundancy in deep transformers. We assume that the input feature $\boldsymbol{H}^\ell$, intermediate vectors $\tilde{\boldsymbol{H}}^\ell$, and the model parameter matrix $\boldsymbol{W}^\ell$ follow normal and independent distributions with mean 0 for all layers. First, we model the growth of the hidden states in a transformer architecture:

**Theorem 2.1** (The growth of the hidden state variance). *Let $\sigma_{\boldsymbol{H}^\ell}^2$ and $\sigma_{\tilde{\boldsymbol{H}}^\ell}^2$ denote the variance of $\boldsymbol{H}^\ell$ and $\tilde{\boldsymbol{H}}^\ell$, respectively. These two variances exhibit the same growth trend, which is*

$$
\Theta(\ell) \le \sigma_{\boldsymbol{H}^\ell}^2 = \sigma_{\boldsymbol{H}^0}^2 \Theta\left(\prod_{k=1}^\ell \left(1 + \frac{1}{\sigma_{\boldsymbol{H}^k}}\right)\right) \le \Theta(\exp(\ell)), \tag{5}
$$

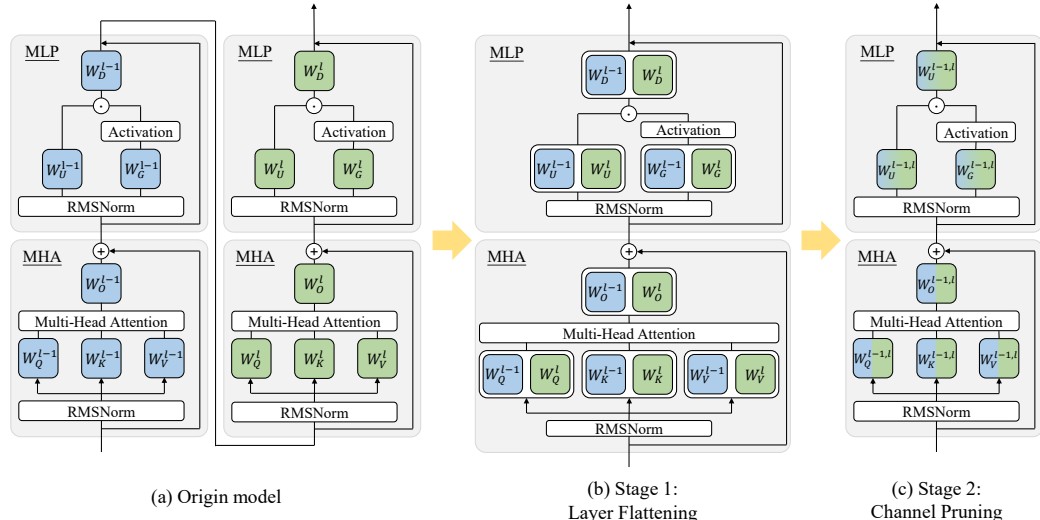

Figure 3: Framework of FlattenGPT. (a) Original stacks of transformer blocks with high similarity. (b) Layer flattening merges two adjacent blocks into one single block with little performance degradation. (c) Flattening bridges the gap between depth compression and channel compression.

This theorem implies that the variance of hidden states could grow as an exponential function of $\ell$. This conclusion is verified by the empirical results in Figure 2(b). Then the following theorem gives the reason why deeper layers are redundant:

**Theorem 2.2** (The norm of gradient). *Let $\frac{\partial y}{\partial \boldsymbol{H}^\ell}$ denote the partial derivative of the model output $y$ to the $\ell$-th hidden states $\boldsymbol{H}^\ell$. The Euclidean norm of this partial derivative is bounded by*

$$\left\| \frac{\partial y}{\partial \boldsymbol{H}^\ell} \right\|_2 \leq \prod_{k=\ell}^{L} \left( 1 + \frac{1}{\sigma_{\boldsymbol{H}^\ell}} A + \frac{1}{\sigma_{\boldsymbol{H}^\ell}^2} B \right), \tag{6}$$

*where $A$ and $B$ are constants for the Transformer network. Specifically, when $\ell = L - c$, where $c$ is a constant number, the limitation of the right-hand side is 1.*

This conclusion implies that for very large $L$, the gradient of deeper layers $x_\ell$, $\left\| \frac{\partial y}{\partial \boldsymbol{H}^\ell} \right\|_2$ is dominated by **identity mapping**, thereby limiting the model's expressivity and hindering its ability to learn meaningful transformations. This conclusion is verified by the empirical results as shown in Figure 2(a), where deeper layers exhibit high cross-layer similarity. The complete proof and more empirical results are given in Appendix A and B, respectively.

Due to this redundancy, previous layer pruning methods delete the entire redundant blocks, *i.e.*, $\text{MHA}^\ell$ or $\text{MLP}^\ell$ (Song et al., 2024; Men et al., 2024). Although these methods achieve promising acceleration as shown in Figure 2(d), pruning at such high granularity will inevitably remove the useful knowledge within the blocks, resulting in a massive performance degradation.

## 3 FLATTENGPT

FlattenGPT strives for a fine-grained parameter removal in depth compression. As illustrated in Figure 3, FlattenGPT employs a two-stage approach to compress the depth in a fine-grained manner. In the first stage, FlattenGPT merges the selected adjacent layers into a single wide layer, flattening the arrangement of layers. Due to the high similarity across layers, the flattening operation hardly alters the inner calculation, reducing the model depth with minimal performance degradation. In the second stage, FlattenGPT adaptively prunes the redundant parameters for the flattened layers, demonstrating less information loss compared with entire layer pruning methods. FlattenGPT produces the same architecture as layer pruning, but preserves important parameters from all layers. It not only runs fast in inference but also maintains high performance, which is a promising way for model compression.

## 3.1 ITERATIVE LAYER FLATTENING

Layer flattening aims to merge layers with high similarity. Since the inputs of the two layers are highly similar, the inner calculation will not be significantly changed by the flattening, therefore preserving better performance. We need to address two issues in this stage: 1) *how to select the layers to flatten*, and 2) *how to merge the selected layers*.

**Layer Selection:** We collect cross-layer feature similarity on a small calibration dataset. Then we design a greedy algorithm to find adjacent layers with the highest similarities iteratively. The algorithm is shown in Algorithm 1. Let $\boldsymbol{S} \in \mathbb{R}^{L \times L}$ denote the similarity matrix, where $\boldsymbol{S}_{i,j}$ denote the cosine similarity between the input feature of layer $i$ and layer $j$. We try to find the two adjacent layers $\{l-1, l\}$ with the highest similarity for each iteration. Then we need to modify the similarity matrix to $\boldsymbol{S}$ for the next iteration. If the next flattened layers $m-1$ and $m$ are not adjacent to the current flattened layers $l-1$ and $l$, there is no problem with directly merging these layers. However, if the next flattened layer are consecutive to the current ones, *i.e.* flattening $\{l-2, l-1, l\}$ (where $m = l - 1$) or $\{l-1, l, l+1\}$ (where $m - 1 = l$), the similarity between the first and the last layer need to be considered. If the input similarity between the first and the last layer is too large, the output of the last layer will be significantly altered, leading to performance degradation. We can modify the similarity matrix $\boldsymbol{S}_{i,j}$ to avoid this problem. For the layers before $l-1$, we remove their similarity with layer $l-1$. Thus, if the flattened layers are consecutive, we can only access their similarity with layer $l$. Similarly, for the layers after $l$, we remove their similarity with the layer $l$. The above steps are iterated until the target number of layers is flattened.

**Layer Flattening:** Flattening aims to merge the selected layers into a single wide layer. To be compatible with the current implementation and AI infrastructure, we construct the flattened layers with the same architecture as the original transformers. Let us denote the target layers as layer $l-1$ and layer $l$. First, we fuse the parameters of affinement in the normalization layer LN with the linear projections in the MHA and MLP. For the MHA layers, we fuse $\boldsymbol{\alpha}_a^{l-1}$ with the query, key, and value matrix:

$$\tilde{\boldsymbol{W}}_{Q,i}^{l-1} = \operatorname{diag}\left(\boldsymbol{\alpha}_a^{l-1}\right) \boldsymbol{W}_{Q,i}^{l-1}, \quad \tilde{\boldsymbol{W}}_{K,i}^{l-1} = \operatorname{diag}\left(\boldsymbol{\alpha}_a^{l-1}\right) \boldsymbol{W}_{K,i}^{l-1}, \quad \tilde{\boldsymbol{W}}_{V,i}^{l-1} = \operatorname{diag}\left(\boldsymbol{\alpha}_a^{l-1}\right) \boldsymbol{W}_{V,i}^{l-1}. \tag{7}$$

For the MLP layers, we fuse $\boldsymbol{\alpha}_p^{l-1}$ with the up and gate matrix:

$$\tilde{\boldsymbol{W}}_u^{l-1} = \operatorname{diag}\left(\boldsymbol{\alpha}_p^{l-1}\right) \boldsymbol{W}_u^{l-1}, \quad \tilde{\boldsymbol{W}}_g^{l-1} = \operatorname{diag}\left(\boldsymbol{\alpha}_p^{l-1}\right) \boldsymbol{W}_g^{l-1}. \tag{8}$$

Similar fuse operations are conducted on layer $l$. After these fusions, the affinement parameters are set to $\boldsymbol{1}_{d_h}$. This step does not change the output of the network but facilitates the next flattening steps.

**MHA flattening** We simply add the output of the two MHA blocks.

$$\begin{aligned}
\operatorname{MHA}^{l-1,l}\left(\boldsymbol{X}\right) &= \sum_{i=1}^{H} \operatorname{Softmax}\left(\sigma_r\left(\boldsymbol{X}\tilde{\boldsymbol{W}}_{Q,i}^{l-1}\right) \sigma_r^{\top}\left(\boldsymbol{X}\tilde{\boldsymbol{W}}_{K,i}^{l-1}\right)\right) \boldsymbol{X}\tilde{\boldsymbol{W}}_{V,i}^{l-1}\boldsymbol{W}_{O,i}^{l-1} \\
&+ \sum_{i=1}^{H} \operatorname{Softmax}\left(\sigma_r\left(\boldsymbol{X}\tilde{\boldsymbol{W}}_{Q,i}^{l}\right) \sigma_r^{\top}\left(\boldsymbol{X}\tilde{\boldsymbol{W}}_{K,i}^{l}\right)\right) \boldsymbol{X}\tilde{\boldsymbol{W}}_{V,i}^{l}\boldsymbol{W}_{O,i}^{l}.
\end{aligned} \tag{9}$$

It is equivalent to concatenating the attention heads from two layers to form an attention block of $2H$ heads. The flattened MHA block can also be calculated by the original implementation, which utilizes a single query, key, value, and output projection matrix.

$$\boldsymbol{W}_Q^{l-1,l} = \left(\tilde{\boldsymbol{W}}_{Q,1}^{l-1}\ \tilde{\boldsymbol{W}}_{Q,2}^{l-1}\ \cdots,\ \tilde{\boldsymbol{W}}_{Q,H}^{l-1}\ \tilde{\boldsymbol{W}}_{Q,1}^{l}\ \tilde{\boldsymbol{W}}_{Q,2}^{l}\ \cdots,\ \tilde{\boldsymbol{W}}_{Q,H}^{l}\right) \tag{10}$$

and similar to the matrix $\tilde{\boldsymbol{W}}_K^{l-1,l}$, $\tilde{\boldsymbol{W}}_V^{l-1,l}$ and $\tilde{\boldsymbol{W}}_O^{l-1,l}$.

**MLP flattening** We simply add the output of two MLP blocks.

$$\operatorname{MLP}^{l-1,l}\left(\boldsymbol{X}\right) = \boldsymbol{X}\tilde{\boldsymbol{W}}_u^{l-1}\sigma_g\left(\boldsymbol{X}\tilde{\boldsymbol{W}}_g^{l-1}\right)\boldsymbol{W}_D^{l-1} + \boldsymbol{X}\tilde{\boldsymbol{W}}_u^{l}\sigma_g\left(\boldsymbol{X}\tilde{\boldsymbol{W}}_g^{l}\right)\boldsymbol{W}_D^{l} \tag{11}$$

It is equivalent to concatenating the hidden states from two layers to form an MLP layer of $2d_{int}$ hidden channels. The full process of layer flattening is shown in Algorithm 1. These steps flatten the layers, which reduces the layers of the original parameters. Flattening does not change the number of parameters and calculations, thus we need a further pruning method to compress the models. We adopt a channel pruning method to compress the parameters in the following part.

---

**Algorithm 1 Iterative layer flattening**

---

**Require:** Base model, number of layers to flatten $N$, calibration set $\mathcal{D}$
1: Calculate the input similarity between each pair of layers $\boldsymbol{S} \in \mathbb{R}^{L \times L}$
2: **while** $N \geq 0$ **do**                                               ▷ Iterative search
3:     Identify the index $(l-1, l)$ of the largest similarity in $\boldsymbol{S}$          ▷ Select adjacent layers
4:     $\tilde{\boldsymbol{W}}_m^j \leftarrow \text{diag}\left(\boldsymbol{\alpha}_a^j\right) \boldsymbol{W}_m^j$, for $m \in \{Q, K, V\}, j \in \{l-1, l\}$    ▷ Fuse affinement parameters in MHA
5:     $\boldsymbol{W}_m^{l-1,l} \leftarrow \left[\tilde{\boldsymbol{W}}_m^{l-1} \; \tilde{\boldsymbol{W}}_m^l\right]$, for $m \in \{Q, K, V\}$          ▷ Flatten MHA
6:     $\boldsymbol{W}_O^{l-1,l} \leftarrow \left[\boldsymbol{W}_O^{l-1,\top} \; \boldsymbol{W}_O^{l,\top}\right]^\top$
7:     $\tilde{\boldsymbol{W}}_m^j \leftarrow \text{diag}\left(\boldsymbol{\alpha}_p^j\right) \boldsymbol{W}_m^j$, for $m \in \{u, g\}, j \in \{l-1, l\}$    ▷ Fuse affinement parameters in MLP
8:     $\boldsymbol{W}_m^{l-1,l} \leftarrow \left[\tilde{\boldsymbol{W}}_m^{l-1} \; \tilde{\boldsymbol{W}}_m^l\right]$, for $m \in \{u, g\}$          ▷ Flatten MLP
9:     $\boldsymbol{W}_D^{l-1,l} \leftarrow \left[\boldsymbol{W}_D^{l-1,\top} \; \boldsymbol{W}_D^{l,\top}\right]^\top$
10:    Delete $\{\boldsymbol{S}_{i,l-1}|i < l\}$ and $\{\boldsymbol{S}_{l,i}|i > l-1\}$ from distance matrix $\boldsymbol{S}$      ▷ Prepare for next iteration
11:    $N \leftarrow N - 1$
12: **end while**

---

## 3.2 Channel Pruning

Layer flattening has produced a high-performance model with fewer layers, and then we need to remove redundant parameters. Previous channel pruning methods are compatible with layer flattening. However, this paper aims to keep the pruned architecture consistent with the original one, thus requiring specific pruning methods. This consistency will simplify the implementation, facilitating the reuse of tuning hyperparameters and deployment. We employ two pruning methods for MHA and MLP blocks, respectively. For the MHA blocks, we prune redundant heads to keep the number of heads and head size unchanged. For the MLP blocks, we prune individual channels with nyström approximation (Gittens & Mahoney, 2016; Musco & Musco, 2017). In this section, we omit the layer index in the formulation for simplicity.

**MHA pruning**   Since the number of heads in the flattened layer is more than the original layer, we aim to prune heads to keep the number of heads the same as the other MHA blocks. We design a metric to compare the importance of head $i$:

$$f_i = \mathbb{E}_{\mathcal{D}}\left[\text{Softmax}\left(\sigma_r\left(\boldsymbol{X}\boldsymbol{W}_{Q,i}\right)\sigma_r^\top\left(\boldsymbol{X}\boldsymbol{W}_{K,i}\right)\right)\boldsymbol{X}\boldsymbol{W}_{V,i} \; \text{diag}\left(\boldsymbol{W}_{O,i}\boldsymbol{W}_{O,i}^\top\right)^{1/2}\right]. \quad (12)$$

This metric estimates the expectation of the attention activation value by multiplying the L2 norm of each line of the output matrix. It measures the impact of the head $i$ on the output of this MHA block. By comparing the impact of each head, we can remove the unimportant heads to prune the MHA block. The complete compression process is shown in Algorithm 2.

---

**Algorithm 2 MHA pruning by removing heads**

---

**Require:** Query, key, value matrix $\boldsymbol{W}_Q, \boldsymbol{W}_K, \boldsymbol{W}_V \in \mathbb{R}^{d_h \times d}$, output matrix $\boldsymbol{W}_O \in \mathbb{R}^{d \times d_h}$, rank $k$, calibration dataset $\mathcal{D}$
1: $f_i = \sum\limits_{i=1}^{N} \text{Softmax}\left(\sigma_r\left(\boldsymbol{X}\boldsymbol{W}_{Q,i}\right)\sigma_r^\top\left(\boldsymbol{X}\boldsymbol{W}_{K,i}\right)\right)\boldsymbol{X}\boldsymbol{W}_{V,i} \; \text{diag}\left(\boldsymbol{W}_{O,i}\boldsymbol{W}_{O,i}^\top\right)^{1/2}$, for $i \in \{1, 2, \cdots, H\}$
2: Let $\boldsymbol{S}_k \in \mathbb{R}^{d \times k}$ be the matrix that selects the top $k$ heads based on $f_i$ scores
3: **return** $(\boldsymbol{W}_Q, \boldsymbol{W}_K, \boldsymbol{W}_V, \boldsymbol{W}_O) \leftarrow \left(\boldsymbol{W}_Q \boldsymbol{S}_k, \boldsymbol{W}_K \boldsymbol{S}_k, \boldsymbol{W}_V \boldsymbol{S}_k, \boldsymbol{S}_k^\top \boldsymbol{W}_O\right)$

---

**MLP pruning**   The MLP blocks conduct important non-linear calculations in the transformer. A simple channel selection is insufficient to maintain the performance of the original model. We employ Nyström approximation (Gittens & Mahoney, 2016; Musco & Musco, 2017) to prune this block. The compression method is shown in Algorithm 3. First, we calculate the ridge leverage score (Musco & Musco, 2017) as the channel importance measurement. Then we select the important channels and adjust the down matrix $\boldsymbol{W}_D$ to compensate for the information loss with Nyström approximation (Gittens & Mahoney, 2016). We have the following theorem which illustrates that Nyström approximation is the best estimation under least squares with L2 regularization. The proof is shown in Appendix A.3.

**Theorem 3.1.** *Let $\boldsymbol{S}_k$ denote a $k$-column selection matrix. Let $\boldsymbol{C}_\sigma$ denote the covariance $\sum_{i=1}^{N} \left(\sigma_g \left(\boldsymbol{X}_i \boldsymbol{W}_U\right)\right)^\top \sigma_s \left(\boldsymbol{X}_i \boldsymbol{W}_U\right)$. The optimal estimation of $\hat{\boldsymbol{W}}_D$ is defined by:*

$$\Delta \hat{\boldsymbol{W}}_D = \underset{\Delta \boldsymbol{W}_D}{\arg\min} \left\| \sigma_s \left(\boldsymbol{X}_i \boldsymbol{W}_U\right) \boldsymbol{S}_k \left(\boldsymbol{S}_k^\top \boldsymbol{W}_D + \Delta \boldsymbol{W}_D\right) - \sigma_s \left(\boldsymbol{X}_i \boldsymbol{W}_U\right) \boldsymbol{W}_D \right\|_2 + \lambda \left\|\Delta \boldsymbol{W}_D\right\|_2,$$

$$\hat{\boldsymbol{W}}_D = \boldsymbol{W}_D + \boldsymbol{S}_k \Delta \hat{\boldsymbol{W}}_D, \tag{13}$$

*where $\lambda$ is the coefficient for L2 regularization. $\Delta \hat{\boldsymbol{W}}_D$ has closed form solution:*

$$\Delta \hat{\boldsymbol{W}}_D = \left(\boldsymbol{S}_k^\top \boldsymbol{C}_\sigma \boldsymbol{S}_k + \lambda \boldsymbol{I}\right)^{-1} \boldsymbol{S}_k^\top \boldsymbol{C}_\sigma \left(\boldsymbol{I} - \boldsymbol{S}_k \boldsymbol{S}_k^\top\right) \boldsymbol{W}_D. \tag{14}$$

---

**Algorithm 3 MLP pruning** by Nyström approximation

---

**Require:** Up and gated $\boldsymbol{W}_u, \boldsymbol{W}_g \in \mathbb{R}^{d_h \times d_{int}}$, down matrix $\boldsymbol{W}_D \in \mathbb{R}^{d_{int} \times d_h}$, rank $k$, calibration dataset $\mathcal{D}$, and ridge intensity $\lambda$
1: Calculate activation correlation $\boldsymbol{C}_\sigma = \sum_{i=1}^{N} \left(\boldsymbol{X}_i \boldsymbol{W}_u \sigma_g \left(\boldsymbol{X}_i \boldsymbol{W}_g\right)\right)^\top \boldsymbol{X}_i \boldsymbol{W}_u \sigma_g \left(\boldsymbol{X}_i \boldsymbol{W}_g\right)$
2: $s_i \leftarrow \left[\boldsymbol{C}_\sigma \left(\boldsymbol{C}_\sigma + \lambda \boldsymbol{I}\right)\right]^{-1}$, for $i \in \{1, 2, \cdots, d_{int}\}$ ▷ Calculate the ridge leverage score
3: Let $\boldsymbol{S}_k \in \mathbb{R}^{d_{int} \times k}$ be the matrix that selects the top $k$ columns based on $s_i$ scores
4: **return** $\left(\boldsymbol{W}_u, \boldsymbol{W}_g, \boldsymbol{W}_D\right) \leftarrow \left(\boldsymbol{W}_u \boldsymbol{S}_k, \boldsymbol{W}_g \boldsymbol{S}_k, \boldsymbol{W}_D + \left(\boldsymbol{S}_k^\top \boldsymbol{C}_\sigma \boldsymbol{S}_k + \lambda \boldsymbol{I}\right)^{-1} \boldsymbol{S}_k^\top \boldsymbol{C}_\sigma \left(\boldsymbol{I} - \boldsymbol{S}_k \boldsymbol{S}_k^\top\right) \boldsymbol{W}_D\right)$

---

### 3.3 Pruning Hyperparameters

All architectural hyperparameters, including width/head count, are predefined to preserve structure consistency with the original transformer block. The target width of the pruned MLP is identical to the original MLP module, and the number of heads is the same as the original MHA module. This setting ensures compatibility with the original AI infrastructure, including GPUs, CUDA kernels, multi-machine communication, inference engine, etc. It is also a clear target for reproducibility.

## 4 Experiments

### 4.1 Experimental Setup

**Models:** We evaluate FlattenGPT on models that employ a sequential transformer block structure: LLaMA-2 (Touvron et al., 2023b), LLaMA-3 (Dubey et al., 2024), Qwen-1.5 (Bai et al., 2023), and Baichuan-2 (Yang et al., 2023), etc. These models share similar architectures of MHA and MLP.

**Implementations and environments:** All hyperparameters, including width/head count, are predefined to preserve structure consistency with the original transformer block. We implement our models using the HuggingFace Transformers library (Wolf et al., 2020). Model compression and performance testing were conducted on 8 NVIDIA A100 80GB GPUs.

**Datasets and Evaluations:** We follow the setup in previous works (Ashkboos et al., 2024; Song et al., 2024) for fairness. The calibration dataset is composed of 128 samples with 2048 tokens, randomly selected from the training split of WikiText-2 (Merity et al., 2016). The evaluation consists of perplexity and zero-shot task performance. The perplexity is evaluated on the test split of WikiText-2 (Merity et al., 2016) dataset. The zero-shot accuracies are evaluated with LM Evaluation Harness (Gao et al., 2024) on Winograd (Sakaguchi et al., 2019), HellaSwag (Zellers et al., 2019), Physical Interaction Question Answering (PIQA) (Bisk et al., 2020), and AI2 Reasoning Challenges (ARC-e, ARC-c) (Clark et al., 2018). We also investigate the effectiveness of recovery finetuning, which employs 50K samples of refined Alpaca (Taori et al., 2023) for instruction tuning with LoRA (Hu et al., 2022). More detailes are presented in Appendix C.

### 4.2 Comparison with Depth Compression

Table 1 shows a comprehensive comparison between FlattenGPT and the other depth compression methods. These methods (Yang et al., 2024; Song et al., 2024; Men et al., 2024; Samragh et al., 2023;

Table 1: Comparison of depth compression methods on WikiText-2 perplexity and zero-shot tasks.

| Model | Method | Sparsity | PPL ↓ | WinoG | HellaS | PIQA | ARC-e | ARC-c | Avg. |
|---|---|---|---|---|---|---|---|---|---|
| LLaMA-2 7B | Dense | 0% | 5.47 | 69.06 | 75.99 | 79.11 | 74.58 | 46.25 | 69.00 |
| | SLEB (Song et al., 2024) | 21.02% | 9.14 | 58.96 | 62.47 | 73.07 | 56.48 | 33.02 | 56.80 |
| | LaCo (Yang et al., 2024) | 21.02% | 50.39 | 60.46 | 54.08 | 68.34 | 55.39 | 35.84 | 54.82 |
| | RM (Samragh et al., 2023) | 21.02% | 676.8 | 49.25 | 29.22 | 54.46 | 34.43 | 22.53 | 37.98 |
| | ShortGPT (Men et al., 2024) | 21.02% | 18.45 | 65.90 | 62.63 | 70.24 | 56.06 | 36.09 | 58.18 |
| | BlockPruner (Zhong et al., 2024) | 21.99% | 11.51 | 62.43 | 65.87 | 74.21 | 61.07 | 37.29 | 60.17 |
| | FlattenGPT | 21.02% | 8.68 | 66.54 | 68.45 | 72.74 | 63.43 | 41.30 | **62.49** |
| LLaMA-2 13B | Dense | 0% | 4.88 | 72.22 | 79.39 | 80.47 | 77.48 | 49.23 | 71.76 |
| | LaCo (Yang et al., 2024) | 24.37% | 13.97 | 59.27 | 60.44 | 72.42 | 54.34 | 34.56 | 56.21 |
| | RM (Samragh et al., 2023) | 24.37% | 10.08 | 66.61 | 66.80 | 73.72 | 66.12 | 41.98 | 63.05 |
| | ShortGPT (Men et al., 2024) | 24.37% | 20.06 | 70.80 | 67.80 | 72.74 | 60.35 | 41.30 | 62.60 |
| | BlockPruner (Zhong et al., 2024) | 25.12% | 8.16 | 66.30 | 72.20 | 76.93 | 65.82 | 41.38 | 64.53 |
| | FlattenGPT | 24.37% | 6.68 | 71.11 | 73.44 | 76.33 | 72.10 | 44.54 | **67.50** |
| Qwen-1.5 7B | Dense | 0% | 7.95 | 66.46 | 76.92 | 79.22 | 62.16 | 42.66 | 65.48 |
| | LaCo (Yang et al., 2024) | 20.97% | 39.23 | 58.64 | 56.35 | 70.40 | 46.89 | 32.85 | 53.03 |
| | RM (Samragh et al., 2023) | 20.97% | 2026 | 49.88 | 42.00 | 67.36 | 54.17 | 28.58 | 48.40 |
| | ShortGPT (Men et al., 2024) | 20.97% | 49.88 | 62.12 | 58.87 | 69.53 | 43.60 | 32.17 | 53.26 |
| | BlockPruner (Zhong et al., 2024) | 21.83% | 20.58 | 55.56 | 59.31 | 71.71 | 53.70 | 33.28 | 54.71 |
| | FlattenGPT | 20.97% | 16.05 | 59.27 | 62.89 | 68.39 | 56.99 | 37.46 | **57.00** |
| Baichuan-2 7B | Dense | 0% | 6.04 | 68.27 | 72.18 | 77.48 | 72.98 | 42.75 | 66.73 |
| | LaCo (Yang et al., 2024) | 21.57% | 26.46 | 58.56 | 51.50 | 68.28 | 52.90 | 28.50 | 51.95 |
| | RM (Samragh et al., 2023) | 21.57% | 189.8 | 52.33 | 30.87 | 59.96 | 38.17 | 23.63 | 40.99 |
| | ShortGPT (Men et al., 2024) | 21.57% | 31.05 | 62.67 | 50.01 | 63.71 | 47.31 | 30.72 | 50.88 |
| | BlockPruner (Zhong et al., 2024) | 22.45% | **15.38** | 61.48 | 58.09 | 69.75 | 58.08 | 33.02 | 56.08 |
| | FlattenGPT | 21.57% | 20.55 | 64.33 | 61.50 | 69.42 | 56.27 | 35.24 | **57.35** |

Table 2: Comparison of pruning methods on throughput, latency, and mean accuracies on zero-shot tasks. Throughput and latency are measured with LLaMA-2 70B on 2 NVIDIA A100 80GB.

| | Method | Sparsity | Throughput (Tokens/s) | Latency (ms) | LLaMA-2 7B | 13B | 70B |
|---|---|---|---|---|---|---|---|
| | Dense | 0% | $299_{1.00\times}$ | $1718.4_{1.00\times}$ | 69.00 | 71.76 | 76.57 |
| 2:4 | SparseGPT | 50% | $293_{0.98\times}$ | $1555.5_{1.10\times}$ | 58.23 | 63.06 | 71.87 |
| | Wanda | 50% | $293_{0.98\times}$ | $1555.5_{1.10\times}$ | 55.59 | 61.23 | 72.34 |
| Width | LLM-Pruner | 20% | $314_{1.05\times}$ | $1534.3_{1.12\times}$ | 62.15 | 67.72 | - |
| | SliceGPT | 20% | $314_{1.05\times}$ | $1658.7_{1.04\times}$ | 58.17 | 63.45 | 72.34 |
| | SliceGPT | 25% | $331_{1.11\times}$ | $1440.7_{1.19\times}$ | 55.49 | 58.90 | 69.75 |
| | SliceGPT | 30% | $343_{1.15\times}$ | $1364.2_{1.26\times}$ | 51.50 | 55.16 | 66.11 |
| Depth | SLEB | 10% | $336_{1.12\times}$ | $1529.1_{1.12\times}$ | 62.24 | 66.77 | 73.14 |
| | SLEB | 20% | $381_{1.27\times}$ | $1364.1_{1.26\times}$ | 56.80 | 62.96 | 70.81 |
| | FlattenGPT | 20% | $\mathbf{381}_{1.27\times}$ | $\mathbf{1364.1}_{1.26\times}$ | 62.49 | 68.27 | 73.94 |

Table 3: Comparison of mean zero-shot accuracies with recovery fine-tuning. The sparsity ratio is 20% and [†] indicates fine-tuned on Alpaca (Taori et al., 2023) dataset.

| | Method | LLaMA-2 7B | LLaMA-2 13B | LLaMA-3 8B |
|---|---|---|---|---|
| | Dense | 69.00 | 71.76 | 73.08 |
| Width | Wanda-sp | 64.53 | 67.37 | - |
| | FLAP | 59.51 | 64.70 | 36.03 |
| | LLM-Pruner | 61.34 | 65.66 | 64.23 |
| | LLM-Pruner[†] | 62.15 | 67.72 | 68.99 |
| Depth | SLEB | 59.25 | 62.96 | - |
| | Shortened LLaMA | 58.36 | 65.86 | 58.30 |
| | Shortened LLaMA[†] | 61.91 | 68.81 | 66.72 |
| | FlattenGPT | 63.83 | 68.27 | 66.21 |
| | FlattenGPT[†] | **66.24** | **70.53** | **70.43** |

Zhong et al., 2024) remove the entire transformer blocks, resulting in massive information loss. Our method alleviates this problem by fine-grained parameter removal and shows superior performance on various model sizes (from 7B to 70B). It achieves the highest perplexity and improves the zero-shot accuracies by at least 2%. FlattenGPT has built a strong approach for depth compression.

## 4.3 COMPARISON WITH OTHER PRUNING METHODS

Table 2 compares the latency, throughput, and mean accuracies on zero-shot tasks of the compressed LLaMA-2 (Touvron et al., 2023b) models. 2:4 pruning methods lead to minor speedup (1.10×) and lower throughput (0.98× with a sparsity ratio of 50%. Width pruning methods, such as SliceGPT (Ashkboos et al., 2024), are more hardware-friendly and speed up the pruned model, while still lagging behind depth pruning methods. FlattenGPT inherits the advantages of acceleration in depth pruning and further improves the performance. Since the compressed model architecture of FlattenGPT is exactly the same as SLEB, the throughput and latency results are the same. FlattenGPT outperforms all other methods in throughput (1.27×), latency (1.26×), and zero-shot tasks performance (about 5% higher), yielding a better trade-off between speed and performance.

## 4.4 RECOVERY FINE-TUNING

The pruned model of FlattenGPT still contains the useful information from all blocks, making it easier for the model to recover performance through Recovery Fine-Tuning (RFT). Table 3 presents the mean accuracies of zero-shot tasks with and without RFT. The results show that the model compressed by FlattenGPT maintains $> 96\%$ zero-shot performance of the dense model, far more than other depth compression methods and some width pruning methods. Even without RFT, our method achieves comparable performance with RFT-based methods. These results illustrate the effectiveness of FlattenGPT.

**More experiments**: We present more experiments and discussions in Appendix D, including additional experiments on various model types and sizes, more pruning methods (van der Ouderaa et al., 2024; Lin et al., 2024), ablation studies, dependency on calibration dataset, and generalization beyond language modeling and transformer architectures. Please kindly refer to this part.

## 5 RELATED WORKS

**Model Pruning** is an approach to compress the number of parameters and calculations in a deep model. Unstructured pruning (LeCun et al., 1989; Hassibi et al., 1993; van der Ouderaa et al., 2024; Dong et al., 2017; Frantar & Alistarh, 2022; 2023; Sun et al., 2023; Zhang et al., 2024b) removes independent weights without pre-determined patterns, leading to sparse weight matrices within the model. This sparsity enables a high pruning ratio but results in complex data access patterns, which are not conducive to hardware acceleration. Structured pruning (Ashkboos et al., 2024; Ma et al., 2023; An et al., 2024) removes elements to form dense matrices that are more efficiently processed by hardware. These methods exhibit a remarkable acceleration but come with worse performance degradation. This paper follows the structured pruning of LLMs, proposing a new depth compression method that balances performance and efficiency.

**Depth compression** aims to reduce the number of layers and speed up inference. Layer pruning approaches use layer importance metrics to remove redundant layers from the model (Men et al., 2024; Samragh et al., 2023; Kim et al., 2024; Song et al., 2024; Zhong et al., 2024; Zhang et al., 2024a). These methods remove the weights and knowledge of the entire layer, limiting the performance of the pruned model. Layer merging methods fuse the parameters of different layers by addition (Yang et al., 2024; Liu et al., 2024; Ding et al., 2025). While this type of method uses information from different layers, simple addition can cause sharp performance degradation. LLM-Streamline (Chen et al., 2025) This paper proposes FlattenGPT, a novel depth compression method that rearranges the layers, which reduces the model depth while retaining the information of each layer and maintains the performance well.

**Width Compression** reduces the number of parameters by reducing the width of the network. LLM-Pruner (Ma et al., 2023) uses gradient magnitudes to estimate the importance of neurons and efficiently fine-tune the performance of the recovery model with parameters. SliceGPT (Ashkboos et al., 2024) and ModeGPT (Lin et al., 2024) employ matrix decomposition to compress the width of each block in the Transformer. The attention head pruning and sharing methods (Michel et al., 2019) were used to reduce the width of the attention module. However, these methods cannot compress model depth, leading to higher inference latency. Our method bridges the gap between channel pruning and layer pruning, which provides a fine-grained layer pruning method and improves the performance.

## 6 CONCLUSION

We propose a novel LLM depth compression method, FlattenGPT, to address the challenges of performance degradation under high-granularity layer pruning. Upon the high similarity of cross-layer input features, we design a layer flattening operation to reduce the model depth with minimal performance loss. Then we adopt channel pruning methods to reduce the number of parameters and calculations in the model. Our proposed method performs well on LLM depth compression, showcasing the effectiveness of fine-grained depth compression. We hope this work can inspire more future efforts in depth compression on neural architectures from the perspective of layer flattening.

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

## A PROOFS

### A.1 PROOF OF THEOREM 2.1

*Proof.* Let $\sigma_{\text{Attn}}$ denote the standard deviation of $\text{Attn}\left(\text{LN}_a^l\left(\boldsymbol{H}^\ell\right)\right)$, and $\sigma_{\text{MLP}}$ denote the standard deviation of $\text{MLP}\left(\text{LN}_p^l\left(\boldsymbol{H}^\ell\right)\right)$. Given equation equation 1 we have:

$$
\begin{aligned}
\text{Var}\left(\tilde{\boldsymbol{H}}^\ell\right) &= \text{Var}\left(\boldsymbol{H}^\ell\right) + \text{Var}\left(\text{Attn}\left(\text{LN}\left(\boldsymbol{H}^\ell\right)\right)\right) + \text{Cov}\left(\text{Attn}\left(\text{LN}\left(\boldsymbol{H}^\ell\right)\right), \boldsymbol{H}^\ell\right) \\
&= \sigma_{\boldsymbol{H}^\ell}^2 + \sigma_{\text{Attn}}^2 + \rho_1 \cdot \sigma_{\boldsymbol{H}^\ell} \cdot \sigma_{\text{Attn}},
\end{aligned}
\tag{15}
$$

$$
\begin{aligned}
\text{Var}\left(\boldsymbol{H}^{\ell+1}\right) &= \text{Var}\left(\tilde{\boldsymbol{H}}^\ell\right) + \text{Var}\left(\text{MLP}\left(\text{LN}\left(\tilde{\boldsymbol{H}}^\ell\right)\right)\right) + \text{Cov}\left(\text{MLP}\left(\text{LN}\left(\tilde{\boldsymbol{H}}^\ell\right)\right), \tilde{\boldsymbol{H}}^\ell\right) \\
&= \sigma_{\tilde{\boldsymbol{H}}^\ell}^2 + \sigma_{\text{MLP}}^2 + \rho_2 \cdot \sigma_{\tilde{\boldsymbol{H}}^\ell} \cdot \sigma_{\text{MLP}},
\end{aligned}
\tag{16}
$$

where $\rho_1, \rho_1$ is the correlation factor. Thus, the evolve from $\text{Var}\left(\boldsymbol{H}^{\ell+1}\right)$ to $\text{Var}\left(\boldsymbol{H}^\ell\right)$ becomes

$$
\sigma_{\boldsymbol{H}^{\ell+1}}^2 = \sigma_{\boldsymbol{H}^\ell}^2 + \sigma_{\text{Attn}}^2 + \sigma_{\text{MLP}}^2 + \rho_1 \cdot \sigma_{\boldsymbol{H}^\ell} \cdot \sigma_{\text{Attn}} + \rho_2 \cdot \sigma_{\tilde{\boldsymbol{H}}^\ell} \cdot \sigma_{\text{MLP}}.
\tag{17}
$$

Let $n$ denote the number of head and $d_{\text{head}}$ denote the dimension of each head. Following results in Sun et al. (2025), based on the independent distribution assumption of weights, we have

$$
\text{Var}\left(\text{Attn}\left(\boldsymbol{Q}, \boldsymbol{k}, \boldsymbol{V}\right)\right) \sim \frac{1}{n}\sum_{i=1}^n d_{\text{head}}\,\text{Var}\left(\boldsymbol{V}_i\right) = \frac{1}{n} \cdot n \cdot \sigma_{\boldsymbol{V}}^2 \cdot d_{\text{head}} = \sigma_{\boldsymbol{W}}^2 d.
\tag{18}
$$

Using the conclusion obtained by Wang et al. (2024), we get

$$
\begin{aligned}
\sigma_{\tilde{\boldsymbol{H}}^\ell}^2 &= \sigma_{\boldsymbol{H}^\ell}^2 + \sigma_{\boldsymbol{W}}^2 + \rho_2 \cdot \sigma_{\boldsymbol{H}^\ell} \cdot \sigma_{\boldsymbol{W}} \\
&= \sigma_{\boldsymbol{H}^\ell}^2 \left(1 + \frac{\sigma_{\boldsymbol{W}}}{\sigma_{\boldsymbol{H}^\ell}} + \rho_2 \cdot \frac{\sigma_{\boldsymbol{W}}^2}{\sigma_{\boldsymbol{H}^\ell}^2}\right) \\
&= \sigma_{\boldsymbol{H}^\ell}^2 \Theta\left(1 + \frac{1}{\sigma_{\boldsymbol{H}^\ell}}\right).
\end{aligned}
\tag{19}
$$

For simplicity, we set the numerator part to 1. Substitute $\sigma_{x'_\ell} = \sigma_{\boldsymbol{H}^\ell}\sqrt{1 + \frac{\sigma_{\boldsymbol{W}}^2}{\sigma_{\boldsymbol{H}^\ell}^2} + \rho_2 \cdot \frac{\sigma_{\boldsymbol{W}}}{\sigma_{\boldsymbol{H}^\ell}}}.$, we can obtain the variance of

$$
\begin{aligned}
\sigma_{\boldsymbol{H}^{\ell+1}}^2 &= \sigma_{\boldsymbol{H}^\ell}^2 + \sigma_{\boldsymbol{W}}^2 + \sigma_{\boldsymbol{W}}^4 d^2 + \rho_1 \cdot \sigma_{\boldsymbol{H}^\ell} \cdot \sigma_{\boldsymbol{W}} + \rho_2 \cdot \sigma_{x'_\ell} \cdot \sigma_{\boldsymbol{W}}^2 d \\
&= \sigma_{\boldsymbol{H}^\ell}^2 + \sigma_{\boldsymbol{W}}^2 + \sigma_{\boldsymbol{W}}^4 d^2 + \rho_1 \cdot \sigma_{\boldsymbol{H}^\ell} \cdot \sigma_{\boldsymbol{W}} + \rho_2 \cdot \sigma_{\boldsymbol{H}^\ell} \cdot \sigma_{\boldsymbol{W}}^2 d + \frac{\rho_2 \sigma_{\boldsymbol{W}}^4 d^2}{2\sigma_{\boldsymbol{H}^\ell}} + \frac{\rho_2^2 \sigma_{\boldsymbol{W}}^3 d\sigma_{\boldsymbol{H}^\ell}}{2} \\
&= \sigma_{\boldsymbol{H}^\ell}^2 \Theta(1 + \frac{1}{\sigma_{\boldsymbol{H}^\ell}}).
\end{aligned}
\tag{20}
$$

The variance with regard to $\sigma_{\boldsymbol{H}}^2$ can be obtained by iteratively apply Equation equation 20:

$$
\sigma_{\boldsymbol{H}^\ell}^2 = \sigma_{\boldsymbol{H}^0}^2 \Theta\left(\prod_{k=1}^{\ell}\left(1 + \frac{1}{\sigma_{\boldsymbol{H}^k}}\right)\right).
\tag{21}
$$

Following the results in Sun et al. (2025), this conclusion could lead to the upper bound and lower bound. Please refer to Appendix A.1 in Sun et al. (2025) for details. $\qquad\square$

## A.2 PROOF OF THEOREM 2.2

*Proof.* For an $L$-layered Pre-LN Transformer, the partial gradient to the $\ell$-th hidden states is given by the chain rule:

$$
\frac{\partial y}{\partial \boldsymbol{H}^\ell} = \prod_{k=\ell}^{L-1}\left(\frac{\partial \boldsymbol{H}^{k+1}}{\partial \tilde{\boldsymbol{H}}^k} \cdot \frac{\partial \tilde{\boldsymbol{H}}^k}{\partial \boldsymbol{H}^k}\right).
\tag{22}
$$

From Sun et al. (2025), we know that

$$
\frac{\partial \boldsymbol{H}^{k+1}}{\partial \tilde{\boldsymbol{H}}^k} \le 1 + \frac{\sigma_{\boldsymbol{W}_U^\ell}\sigma_{\boldsymbol{W}_D^\ell}}{\sigma_{\tilde{\boldsymbol{H}}^k}(\sqrt{d} + \sqrt{d_{\text{MLP}}})^2} = 1 + \frac{\sigma_\ell^2}{\sigma_{\tilde{\boldsymbol{H}}^k}(\sqrt{d} + \sqrt{d_{\text{MLP}}})^2}.
\tag{23}
$$

From Papaspiliopoulos (2020), we get

$$
\frac{\partial \tilde{\boldsymbol{H}}^k}{\partial \boldsymbol{H}^k} \le \left(1 + 2dh\left(\sqrt{s} + 2 + \frac{1}{\sqrt{s}}\right)\frac{\sigma^2}{\sigma_{\boldsymbol{H}^\ell}}\left(\sigma^2 d\sqrt{d_{\text{head}}} + \left(1 + \sqrt{d_{\text{head}}/d}\right)\right)\right),
\tag{24}
$$

where $h$ denotes the number of heads and $s$ denotes the sequence length, respectively. Following the proof from Sun et al. (2025), the target equation can be expressed as

$$
\left\|\frac{\partial y}{\partial \boldsymbol{H}^\ell}\right\|_2 \le \prod_{k=\ell}^{L}\left(1 + \frac{1}{\sigma_{\boldsymbol{H}^k}}A + \frac{1}{\sigma_{\boldsymbol{H}^k}^2}B\right),
\tag{25}
$$

where

$$
A = \frac{\sigma^2}{(\sqrt{d} + \sqrt{d_{\text{FFN}}})^2} + 2dh\left(\sqrt{s} + 2 + \frac{1}{\sqrt{s}}\right)\sigma^2\left(d\sqrt{d_{\text{head}}} + 1 + \sqrt{d_{\text{head}}/d}\right),
\tag{26}
$$

$$
B = 2dh\left(\sqrt{s} + 2 + \frac{1}{\sqrt{s}}\right)\sigma^4 d\sqrt{d_{\text{head}}}.
\tag{27}
$$

This conclusion indicates that for the deep layers in the model, the partial gradient $\frac{\partial y}{\partial \boldsymbol{H}^\ell}$ will be bounded. Considering the exponential growth in Theorem 2.1, the partial gradient will be bounded by

$$
\begin{aligned}
\left\|\frac{\partial y}{\partial \boldsymbol{H}^\ell}\right\|_2 &\le \prod_{k=\ell}^{L}\left(1 + \frac{1}{\sigma_{\boldsymbol{H}^k}}A + \frac{1}{\sigma_{\boldsymbol{H}^k}^2}B\right) \\
&\le \prod_{k=\ell}^{L}\left(1 + \frac{1}{\sigma_{\boldsymbol{H}^\ell}}A + \frac{1}{\sigma_{\boldsymbol{H}^\ell}^2}B\right) \\
&= \left(1 + \frac{1}{\sigma_{\boldsymbol{H}^\ell}}A + \frac{1}{\sigma_{\boldsymbol{H}^\ell}^2}B\right)^{L-\ell}.
\end{aligned}
\tag{28}
$$

Let $L - \ell$ be a constant number $c$, which implies the $c$-th layer from the last. As $\ell$ grows, $\sigma_{\boldsymbol{H}^\ell}$ will grow to infinity. Then we get

$$\lim_{\ell \to +\infty} \left\| \frac{\partial y}{\partial \boldsymbol{H}^\ell} \right\|_2 \leq \lim_{\ell \to +\infty} \left( 1 + \frac{1}{\sigma_{\boldsymbol{H}^\ell}} A + \frac{1}{\sigma_{\boldsymbol{H}^\ell}^2} B \right)^c = 1. \tag{29}$$

$\square$

### A.3 PROOFS OF THEOREM 3.1

*Proof.* The original solution for linear regression with L2 regularization is defined as follows:

**Lemma A.1.** *Let $\boldsymbol{X} \in \mathbb{R}^{n \times p}$ be the design matrix, $\boldsymbol{y} \in \mathbb{R}^n$ be the response vector, and $\lambda > 0$ be the regularization parameter. The L2 regularized linear regression (Ridge Regression) minimizes the following objective function:*

$$\arg\min_{\boldsymbol{\theta}} \|\boldsymbol{X}\boldsymbol{\theta} - \boldsymbol{y}\|_2^2 + \lambda \|\boldsymbol{\theta}\|_2^2, \tag{30}$$

*where $\boldsymbol{\theta} \in \mathbb{R}^p$ is the coefficient vector. The closed-form solution for the optimal coefficient vector $\hat{\boldsymbol{\theta}}$ is given by:*

$$\hat{\boldsymbol{\theta}} = (\boldsymbol{X}^\top \boldsymbol{X} + \lambda \boldsymbol{I})^{-1} \boldsymbol{X}^\top \boldsymbol{y}, \tag{31}$$

*provided that the matrix $(\boldsymbol{X}^\top \boldsymbol{X} + \lambda \boldsymbol{I})$ is invertible. Here, $\boldsymbol{I}$ denotes the $p \times p$ identity matrix. This solution always exists for $\lambda > 0$, even when $\boldsymbol{X}^\top \boldsymbol{X}$ is singular.*

The proof of this lemma can be found in most linear algebra textbooks (Montgomery et al., 2021). As for Equation equation 13, substitute $\theta = \Delta \boldsymbol{W}_D$, $\boldsymbol{X} = \sigma_s(\boldsymbol{X}_i \boldsymbol{W}_U) \boldsymbol{S}_k$, $\boldsymbol{y} = \sigma_s(\boldsymbol{X}_i \boldsymbol{W}_U) (\boldsymbol{I} - \boldsymbol{S}_k \boldsymbol{S}_k^\top) \boldsymbol{W}_D$, and then we will the solutions. $\square$

## B EMPIRICAL RESULTS

We present more empirical results in various architectures, including LLaMA-2 (Touvron et al., 2023b) at {7B, 13B}, Qwen-1.5 (Bai et al., 2023) at {7B, 14B}, and Baichuan-2 at {7B, 13B} (Yang et al., 2023). As shown in Figure 4, the high cross-layer similarity and large feature norm is consistent across various model types and parameter sizes. According to the theoretical analysis above, this phenomenon is deeply related to the architecture of transformers.

## C IMPLEMENTATIONS

### C.1 MODIFIED ALGORITHMS FOR GROUPED QUERY ATTENTION

Some modern LLMs, such as LLaMA-3 (Dubey et al., 2024), utilize a shared key-value strategy to improve inference efficiency, which is denoted as Grouped Query Attention (GQA). To keep the pruned architecture the same as the original attention blocks, we modify the channel pruning on MHA. Instead of finding the least important attention head individually, we find the least important pair of key and value. Then we delete this pair and its corresponding queries. We apply this modification to LLaMA-3 8B compression in the paper.

### C.2 IMPLEMENTATION DETAILS

**Setup** We utilize the HuggingFace generation library (Wolf et al., 2020) to implement our LLM models and use PyTorch (Paszke et al., 2019) Hooks for hidden states recording and correlation matrix estimations. Unless otherwise specified, the experiments were conducted on 8 NVIDIA H800 80GB GPUs. The models use the BF16 data format. The calibration set consists of a random sample of 128 sequences, each of length 2048, from WikiText-2, following the common practice in the literature (Ashkboos et al., 2024).

**Datasets** We consider multiple tasks in LM Evaluation Harness (Gao et al., 2024), including ARC-e, ARC-c (Clark et al., 2018), PIQA (Bisk et al., 2020), WinoGrande (Sakaguchi et al., 2019), and HellaSwag (Zellers et al., 2019).

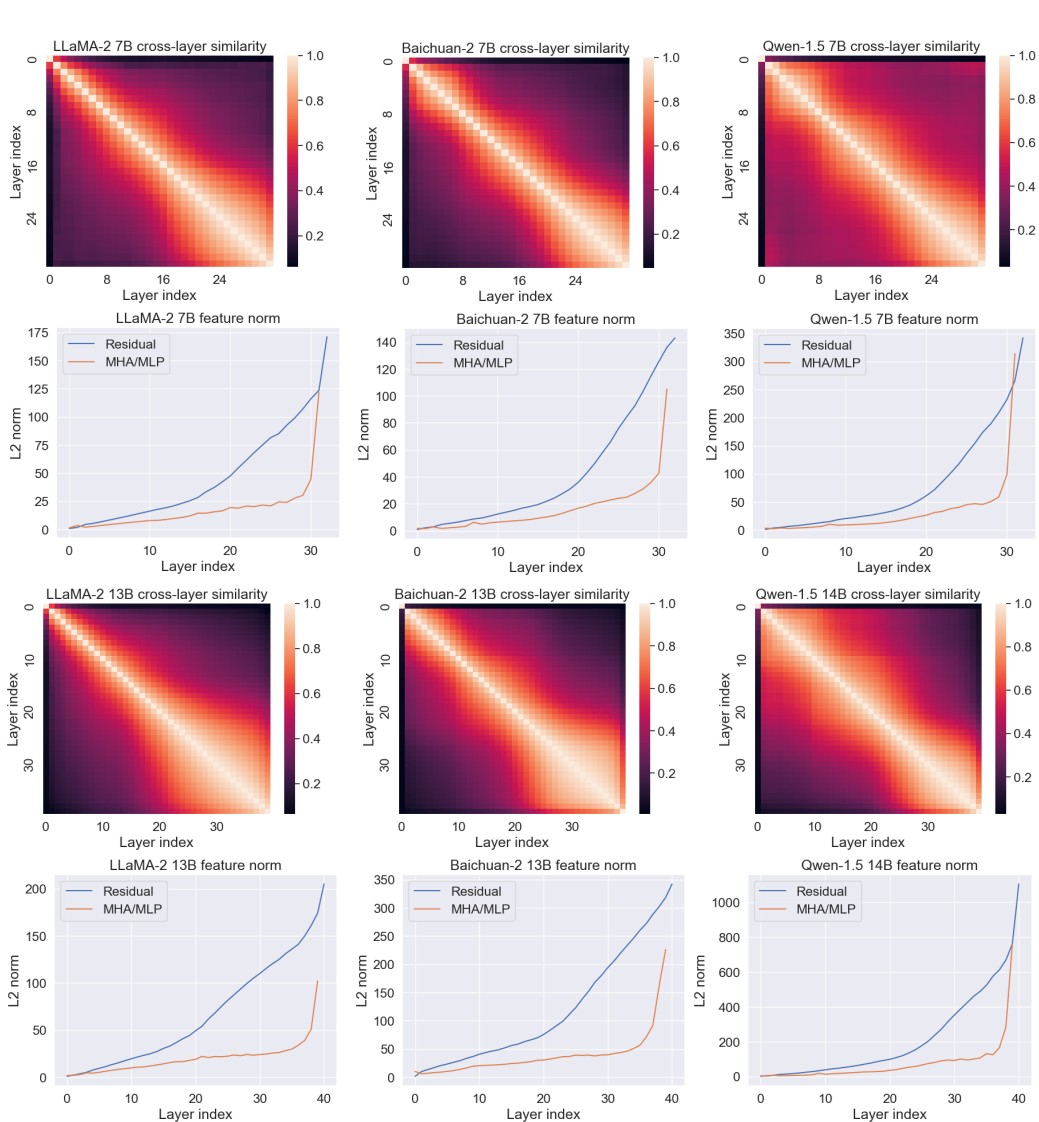

Figure 4: Cross-layer similarity and feature norm in multiple architectures.

**Correlation Matrix Estimations**   Our algorithms utilize input correlation matrices in the MLP pruning method. We gather the empirical data from the calibration set by registering the PyTorch (Paszke et al., 2019) hooks in the model. Our process compresses the MLP blocks from all layers first, then compresses the MHA blocks from all layers.

**Matrix Operations**   We utilize 'torch.linalg.solve' in PyTorch for computing the inverse on tensors of dtype FP32.

**MLP module**   Our MLP pruning method requires a ridge leverage score parameter $\lambda$. We set $\lambda$ to 10 times the mean singular value of the correlation matrix across all experiments.

**MHA module**   Our MHA pruning method removes the entire group of query, key, and value matrices.

**Recovery Fine-Tuning**   We use 50K samples of refined Alpaca for instruction tuning. The learning rate is set to $1 \times 10^{-4}$. The other primary hyperparameters used are $lora\_alpha = 16$, $lora\_r = 8$, and $batch\_size = 64$.

**Latency and Throughput**   We test the throughput and latency results on token generation and prompt processing, respectively. For token generation, we generate sentences with a length of 128 tokens and a batch size of 64. For prompt processing, we measure the latency when processing an input sequence with 2048 tokens.

## C.3   THROUGHPUT AND LATENCY

Table 5 compares the latency, throughput, and perplexities of the compressed LLaMA-2 (Touvron et al., 2023b) models with other pruning methods. We test the throughput and latency results on token generation and prompt processing, respectively. For token generation, we generate sentences with a length of 128 tokens and a batch size of 64. For prompt processing, we measure the latency when processing an input sequence with 2048 tokens. Even with dedicated hardware support, 2:4 pruning methods still lead to minor speedup ($1.10\times$) and lower throughput ($0.98\times$ with a sparsity ratio of 50%. Width pruning methods, such as SliceGPT (Ashkboos et al., 2024), are more hardware-friendly and speed up the pruned model, while still lagging behind depth pruning methods. FlattenGPT inherits the advantages of acceleration in depth pruning and further improves the performance. Since the compressed model architecture of FlattenGPT is exactly the same as SLEB, the throughput and latency results are the same. FlattenGPT outperforms all other methods in throughput ($1.27\times$) and latency ($1.26\times$), and achieves a comparable perplexities. These results demonstrate that FlattenGPT has a better trade-off between speed and performance.

## C.4   PRUNING COMPUTATION COST

Table 6 compares the compression times of FlattenGPT with the prevailing pruning methods, including SliceGPT (Ashkboos et al., 2024), LLM surgeon (van der Ouderaa et al., 2024), and MoDeGPT (Lin et al., 2024). LLM Surgeon requires the gradient information of the LLMs, leading to heavy computation. SliceGPT and ModeGPT do not leverage gradients, they can compress a model with fewer GPUs and computation time. Our approach, FlattenGPT, is even faster than these methods, as we collect the correlation matrix of all layers at the same time. Thus FlattenGPT is an efficient pruning method in this area.

## D   EXPERIMENTS

## D.1   ADDITIONAL COMPARISON OF TRAINING-FREE PRUNING METHODS

We compare the performance with other training-free pruning methods in Table 4, including both width compression and depth compression. The width compression includes the 2:4 pruning methods SparseGPT (Frantar & Alistarh, 2023) and Wanda (Sun et al., 2023), and structured channel pruning methods SliceGPT (Ashkboos et al., 2024). The depth compression includes LaCo (Yang et al.,

Table 4: Comparison with training-free pruning methods on WikiText-2 perplexity and accuracies on zero-shot tasks.

| | Method | Sparsity | PPL ↓ | WinoG | HellaS | PIQA | ARC-e | ARC-c | Avg. |
|---|---|---|---|---|---|---|---|---|---|
| | LLaMA-2 7B (original) | 0% | 5.47 | 69.06 | 75.99 | 79.11 | 74.58 | 46.25 | 69.00 |
| Width | SpareGPT (Frantar & Alistarh, 2023) | 2:4 (50%) | 10.79 | 64.96 | 58.93 | 72.14 | 60.90 | 34.22 | 58.23 |
| | Wanda (Sun et al., 2023) | 2:4 (50%) | 12.09 | 62.27 | 55.33 | 70.84 | 57.58 | 31.91 | 55.59 |
| | SliceGPT (Ashkboos et al., 2024) | 21.45% | 7.02 | 59.91 | 56.04 | 72.42 | 63.64 | 37.12 | 57.83 |
| Depth | SLEB (Song et al., 2024) | 21.02% | 9.14 | 58.96 | 62.47 | 73.07 | 56.48 | 33.02 | 56.80 |
| | LaCo (Yang et al., 2024) | 21.02% | 50.39 | 60.46 | 54.08 | 68.34 | 55.39 | 35.84 | 54.82 |
| | RM (Samragh et al., 2023) | 21.02% | 676.8 | 49.25 | 29.22 | 54.46 | 34.43 | 22.53 | 37.98 |
| | ShortGPT (Men et al., 2024) | 21.02% | 18.45 | 65.90 | 62.63 | 70.24 | 56.06 | 36.09 | 58.18 |
| | BlockPruner (Zhong et al., 2024) | 21.99% | 11.51 | 62.43 | 65.87 | 74.21 | 61.07 | 37.29 | 60.17 |
| | FlattenGPT | 21.02% | **8.68** | 66.54 | 68.45 | 72.74 | 63.43 | 41.30 | **62.49** |
| | LLaMA-2 13B (original) | 0% | 4.88 | 72.22 | 79.39 | 80.47 | 77.48 | 49.23 | 71.76 |
| Width | SpareGPT (Frantar & Alistarh, 2023) | 2:4 (50%) | 8.75 | 68.51 | 65.52 | 75.46 | 66.04 | 39.76 | 63.06 |
| | Wanda (Sun et al., 2023) | 2:4 (50%) | 8.99 | 67.01 | 63.09 | 73.94 | 64.31 | 37.80 | 61.23 |
| | SliceGPT (Ashkboos et al., 2024) | 25% | 6.63 | 67.48 | 58.10 | 68.55 | 62.50 | 37.88 | 58.90 |
| Depth | LaCo (Yang et al., 2024) | 24.37% | 13.97 | 59.27 | 60.44 | 72.42 | 54.34 | 34.56 | 56.21 |
| | RM (Samragh et al., 2023) | 24.37% | 10.08 | 66.61 | 66.80 | 73.72 | 66.12 | 41.98 | 63.05 |
| | ShortGPT (Men et al., 2024) | 24.37% | 20.06 | 70.80 | 67.80 | 72.74 | 60.35 | 41.30 | 62.60 |
| | BlockPruner (Zhong et al., 2024) | 25.12% | 8.16 | 66.30 | 72.20 | 76.93 | 65.82 | 41.38 | 64.53 |
| | FlattenGPT | 24.37% | **6.68** | 71.11 | 73.44 | 76.33 | 72.10 | 44.54 | **67.50** |
| | LLaMA-2 70B (original) | 0% | 3.32 | 77.98 | 83.84 | 82.70 | 80.98 | 57.34 | 76.57 |
| Width | SpareGPT (Frantar & Alistarh, 2023) | 2:4 (50%) | 5.70 | 76.56 | 76.09 | 80.03 | 76.94 | 49.74 | 71.87 |
| | Wanda Sun et al. (2023) | 2:4 (50%) | 5.48 | 74.66 | 79.22 | 80.30 | 76.35 | 51.19 | 72.34 |
| | SliceGPT Ashkboos et al. (2024) | 20% | 4.44 | 74.92 | 72.98 | 76.61 | 80.51 | 55.20 | 72.34 |
| Depth | SLEB Song et al. (2024) | 19.84% | 4.88 | 72.93 | 77.21 | 80.14 | 75.38 | 48.38 | 70.81 |
| | ShortGPT Ashkboos et al. (2024) | 19.84% | 66.33 | 71.96 | 77.62 | 76.02 | 76.02 | 52.95 | 71.68 |
| | FlattenGPT | 19.84% | **4.79** | 77.35 | 81.42 | 80.36 | 77.48 | 53.07 | **73.94** |
| | Baichuan-2 7B (original) | 0% | 6.04 | 68.27 | 72.18 | 77.48 | 72.98 | 42.75 | 66.73 |
| Depth | LaCo (Yang et al., 2024) | 21.57% | 26.46 | 58.56 | 51.50 | 68.28 | 52.90 | 28.50 | 51.95 |
| | RM (Samragh et al., 2023) | 21.57% | 189.8 | 52.33 | 30.87 | 59.96 | 38.17 | 23.63 | 40.99 |
| | ShortGPT (Men et al., 2024) | 21.57% | 31.05 | 62.67 | 50.01 | 63.71 | 47.31 | 30.72 | 50.88 |
| | BlockPruner (Zhong et al., 2024) | 22.45% | **15.38** | 61.48 | 58.09 | 69.75 | 58.08 | 33.02 | 56.08 |
| | FlattenGPT | 21.57% | 20.55 | 64.33 | 61.50 | 69.42 | 56.27 | 35.24 | **57.35** |
| | Baichuan-2 13B (original) | 0% | 6.66 | 70.40 | 75.23 | 78.84 | 74.07 | 47.70 | 69.25 |
| Depth | LaCo (Yang et al., 2024) | 22.68% | 27.07 | 58.01 | 54.00 | 70.89 | 57.11 | 32.94 | 54.59 |
| | RM (Samragh et al., 2023) | 22.68% | 17.70 | 67.88 | 63.78 | 68.99 | 57.49 | 37.54 | 59.14 |
| | ShortGPT (Men et al., 2024) | 22.68% | 20.69 | 68.27 | 61.71 | 69.31 | 56.52 | 36.69 | 58.50 |
| | BlockPruner (Zhong et al., 2024) | 24.19% | 15.36 | 64.01 | 64.20 | 71.44 | 59.81 | 37.88 | 59.47 |
| | FlattenGPT | 22.68% | **13.71** | 68.19 | 65.27 | 71.22 | 58.75 | 37.03 | **60.09** |
| | Qwen-1.5 7B (original) | 0% | 7.95 | 66.46 | 76.92 | 79.22 | 62.16 | 42.66 | 65.48 |
| Depth | LaCo (Yang et al., 2024) | 20.97% | 39.23 | 58.64 | 56.35 | 70.40 | 46.89 | 32.85 | 53.03 |
| | RM (Samragh et al., 2023) | 20.97% | 2026 | 49.88 | 42.00 | 67.36 | 54.17 | 28.58 | 48.40 |
| | ShortGPT (Men et al., 2024) | 20.97% | 49.88 | 62.12 | 58.87 | 69.53 | 43.60 | 32.17 | 53.26 |
| | BlockPruner (Zhong et al., 2024) | 21.83% | 20.58 | 55.56 | 59.31 | 71.71 | 53.70 | 33.28 | 54.71 |
| | FlattenGPT | 20.97% | **16.05** | 59.27 | 62.89 | 68.39 | 56.99 | 37.46 | **57.00** |
| | Qwen-1.5 14B (original) | 0% | 7.44 | 70.56 | 79.41 | 79.87 | 68.48 | 47.01 | 69.07 |
| Depth | LaCo (Yang et al., 2024) | 22.25% | 16.32 | 58.33 | 60.16 | 71.55 | 53.70 | 34.04 | 55.56 |
| | RM (Samragh et al., 2023) | 22.25% | 55.99 | 53.28 | 42.08 | 67.08 | 50.72 | 29.01 | 48.43 |
| | ShortGPT (Men et al., 2024) | 22.25% | 1237 | 55.96 | 36.16 | 58.60 | 38.09 | 34.81 | 44.72 |
| | BlockPruner (Zhong et al., 2024) | 23.72% | 15.67 | 61.48 | 66.92 | 75.24 | 59.51 | 39.08 | 60.45 |
| | FlattenGPT | 22.25% | **11.55** | 65.59 | 68.57 | 74.10 | 65.03 | 40.78 | **62.81** |

2024), SLEB (Song et al., 2024), Relative magnitude (Samragh et al., 2023), ShortGPT (Men et al., 2024), and BlockPruner (Zhong et al., 2024). FlattenGPT outperforms these methods on WikiText-2 perplexity and accuracy on the zero-shot downstream tasks, showcasing the effectiveness of our method.

## D.2 ADDITIONAL COMPARISON OF RECOVERY FINE-TUNING

Table 7 shows the impact of Recovery Fine-Tuning (RFT). Our method outperforms previous methods after RFT. This is because the flattening method retains the knowledge from all layers and makes it easier for fine-tuning.

Table 5: Throughput (tokens/s), latency (ms), and perplexity on WikiText-2 test split results. Throughput and latency are measured with LLaMA-2-70B on 2 NVIDIA A100 GPUs.

| Method | Pruning Unit | Sparsity | Throughput (Tokens/s) | Improve ↑ | Latency (ms) | Speedup ↑ | LLaMA-2 7B | 13B | 70B |
|---|---|---|---|---|---|---|---|---|---|
| Dense | - | 0% | 299 | 1.00× | 1718.4 | 1.00× | 5.47 | 4.88 | 3.32 |
| SparseGPT | 2:4 | 50% | 293 | 0.98× | 1555.5 | 1.10× | 10.79 | 8.75 | 5.70 |
| Wanda | 2:4 | 50% | 293 | 0.98× | 1555.5 | 1.10× | 12.09 | 8.99 | 5.48 |
| DSnoT | 2:4 | 50% | 293 | 0.98× | 1555.5 | 1.10× | 11.97 | 8.87 | 5.49 |
| LLM-Pruner | Width | 20% | 314 | 1.05× | 1534.3 | 1.12× | 10.58 | 8.56 | - |
| SliceGPT | Width | 20% | 314 | 1.05× | 1658.7 | 1.04× | 6.87 | 6.01 | 4.44 |
| SliceGPT | Width | 25% | 331 | 1.11× | 1440.7 | 1.19× | 7.55 | 6.63 | 4.89 |
| SliceGPT | Width | 30% | 343 | 1.15× | 1364.2 | 1.26× | 8.59 | 7.44 | 5.44 |
| SLEB | Depth | 20% | 381 | 1.27× | 1364.1 | 1.26× | 9.14 | 6.80 | 4.88 |
| FlattenGPT | Depth | 20% | 381 | 1.27× | 1364.1 | 1.26× | 8.68 | 6.50 | 4.79 |

Table 6: Computation cost of pruning 20% with FlattenGPT and recovery fine-tuning on a NVIDIA H800 80GB. The calibration dataset consists of 128 samples with a sequence length of 2048.

| Method | Model | Pruning Time | Pruning GPUs | RFT Time | RFT GPUs | Total |
|---|---|---|---|---|---|---|
| SliceGPT | LLaMA-2 7B | 44m | 1 H100 80GB | 23m | 1 H100 80GB | 1h07m |
| | LLaMA-2 13B | 1h08m | 1 H100 80GB | 44m | 1 H100 80GB | 1h52m |
| LLM surgeon | LLaMA-2 7B | 17h08m | 4 H100 80GB | - | - | - |
| | LLaMA-2 13B | 1d9h26m | 8 H100 80GB | - | - | - |
| ModeGPT | LLaMA-2 7B | 4h09m | 1 A100 80GB | 31m | 1 A100 80GB | 4h40m |
| | LLaMA-2 13B | 8h26m | 1 A100 80GB | - | - | - |
| FlattenGPT | LLaMA-2 7B | 7m | 1 H800 80GB | 25m | 1 H800 80GB | 32m |
| | LLaMA-2 13B | 24m | 1 H800 80GB | 45m | 1 H800 80GB | 1h09m |

## D.3 EFFECTIVENESS OF FLATTENING

**Flattened layer indices:** We show which transformer blocks are chosen to be flattened in Figure 5. The location of flattened transformer blocks is highly consistent across various target models. The late blocks are almost flattened except the last one or two, whereas the early blocks are rarely selected. This is related to the similarity distribution in the model, where the late blocks have more similar input features.

**Performance after flattening:** We need to answer the question: *How does flattening improve the performance of the depth-compressed model?* The answer is that **Flattening preserves more knowledge**. Compared with the layer pruning methods, flattening preserves the parameters and thus preserves the knowledge in the parameters. This knowledge facilitates performance maintenance during depth compression. Figure 6 illustrates the comparison of layer pruning and layer flattening on LLaMA-2 7B. We use the same layer index in both settings, *i.e.*, to prune the selected layer or merge the selected layer with the prior layer. In the flattening experiments, the model performance gradually drops as the number of flattened layers increases. After flattening 8 layers, it has maintained 98% of accuracy on zero-shot tasks and has a 19% degradation on perplexity. This result leaves plenty of room for channel pruning. However, on the contrary, layer pruning quickly loses performance with merely one or two pruned layers. It only maintains 80% of accuracy on zero-shot tasks and 319% degradation on perplexity! With such information loss, layer-pruning-based methods are very limited and cannot achieve high performance. Our flattening method has alleviated this problem, thus providing an effective way of depth compression.

## D.4 EFFECTIVENESS OF OUR CHANNEL PRUNING METHOD

The flattening operation changes the depth compression task into a channel pruning task. This method shows an advantage of fine-grained depth compression, whereas it relies on the performance of the channel pruning method. In this paper, we use a simple yet effective channel pruning method. To validate the effectiveness of our channel pruning method, we conduct experiments with channel pruning only. We use the sparsity distribution described in ModeGPT (Lin et al., 2024), and compare the channel pruning performance with other channel pruning methods. As shown in Table 8, our

Table 7: Zero-shot task performance of recovery fine-tuning. [†] indicates fine-tuned on Alpaca (Taori et al., 2023) dataset.

| | Method | Sparsity | WinoG | HellaS | PIQA | ARC-e | ARC-c | Avg. |
|---|---|---|---|---|---|---|---|---|
| | LLaMA-2 7B (original) | 0% | 69.06 | 75.99 | 79.11 | 74.58 | 46.25 | 69.00 |
| Width | Wanda-sp (Sun et al., 2023) | 18.81% | 63.77 | 70.66 | 76.44 | 69.61 | 42.15 | 64.53 |
| | FLAP (An et al., 2024) | 19.19% | 64.72 | 64.69 | 73.39 | 62.25 | 32.51 | 59.51 |
| | LLM-Pruner (Ma et al., 2023) | 18.82% | 61.17 | 66.13 | 76.66 | 64.86 | 37.88 | 61.34 |
| | LLM-Pruner[†] (Ma et al., 2023) | 18.82% | 61.88 | 67.13 | 77.48 | 65.78 | 38.48 | 62.15 |
| Depth | SLEB (Song et al., 2024) | 18.02% | 59.75 | 63.95 | 73.94 | 63.47 | 35.15 | 59.25 |
| | Shortened LLaMA (Kim et al., 2024) | 18.02% | 57.46 | 63.36 | 73.78 | 64.02 | 33.19 | 58.36 |
| | Shortened LLaMA[†] (Kim et al., 2024) | 18.02% | 58.80 | 67.99 | 76.06 | 68.81 | 37.88 | 61.91 |
| | SLM (Ding et al., 2025) | 18.02% | 66.30 | 65.10 | 70.24 | 61.45 | 38.31 | 60.28 |
| | SLM[†] (Ding et al., 2025) | 18.02% | 67.09 | 70.48 | 73.67 | 69.11 | 41.21 | 64.31 |
| | FlattenGPT | 18.02% | 67.40 | 70.74 | 74.59 | 64.44 | 41.98 | 63.83 |
| | FlattenGPT[†] | 18.02% | 68.75 | 73.01 | 74.97 | 67.40 | 45.05 | 66.24 |
| | LLaMA-2 13B (original) | 0% | 72.22 | 79.39 | 80.47 | 77.48 | 49.23 | 71.76 |
| Width | Wanda-sp (Sun et al., 2023) | 19.49% | 67.01 | 74.75 | 77.48 | 73.48 | 44.11 | 67.37 |
| | FLAP (An et al., 2024) | 19.47% | 68.35 | 69.07 | 74.65 | 70.83 | 40.61 | 64.70 |
| | LLM-Pruner (Ma et al., 2023) | 19.48% | 64.17 | 72.02 | 78.51 | 69.99 | 43.60 | 65.66 |
| | LLM-Pruner[†] (Ma et al., 2023) | 19.48% | 67.32 | 74.84 | 79.16 | 73.49 | 43.77 | 67.72 |
| Depth | SLEB (Song et al., 2024) | 19.50% | 64.96 | 70.55 | 76.61 | 64.35 | 38.31 | 62.96 |
| | Shortened LLaMA (Kim et al., 2024) | 19.50% | 70.48 | 71.19 | 75.03 | 69.53 | 43.09 | 65.86 |
| | Shortened LLaMA[†] (Kim et al., 2024) | 19.50% | 71.11 | 75.20 | 76.28 | 74.79 | 46.67 | 68.81 |
| | SLM (Ding et al., 2025) | 19.50% | 70.80 | 67.73 | 72.36 | 64.82 | 39.68 | 63.08 |
| | SLM[†] (Ding et al., 2025) | 19.50% | 71.67 | 76.37 | 77.42 | 76.56 | 48.55 | 70.11 |
| | FlattenGPT | 19.50% | 71.43 | 75.26 | 77.58 | 71.68 | 45.39 | 68.27 |
| | FlattenGPT[†] | 19.50% | 71.82 | 77.85 | 78.73 | 75.08 | 49.15 | 70.53 |
| | LLaMA-3 8B (original) | 0% | 73.40 | 79.17 | 79.49 | 80.09 | 53.24 | 73.08 |
| Width | FLAP (An et al., 2024) | 16.30% | 49.96 | 26.36 | 52.18 | 26.81 | 24.83 | 36.03 |
| | LLM-Pruner (Ma et al., 2023) | 15.39% | 68.67 | 67.79 | 77.04 | 68.60 | 39.08 | 64.23 |
| | LLM-Pruner[†] (Ma et al., 2023) | 15.39% | 70.32 | 74.27 | 79.49 | 74.29 | 46.59 | 68.99 |
| Depth | Shortened LLaMA (Kim et al., 2024) | 16.30% | 57.85 | 60.99 | 73.23 | 65.40 | 34.04 | 58.30 |
| | Shortened LLaMA[†] (Kim et al., 2024) | 16.30% | 62.75 | 72.70 | 78.07 | 75.30 | 44.80 | 66.72 |
| | SLM (Ding et al., 2025) | 16.30% | 69.61 | 61.8 | 71.98 | 66.04 | 41.81 | 62.25 |
| | SLM[†] (Ding et al., 2025) | 16.30% | 71.74 | 73.77 | 77.64 | 76.60 | 50.94 | 70.14 |
| | FlattenGPT | 16.30% | 71.82 | 70.63 | 72.91 | 69.1 | 46.59 | 66.21 |
| | FlattenGPT[†] | 16.30% | 73.09 | 75.93 | 77.09 | 75.72 | 50.34 | 70.43 |

Table 8: Zero-shot task performance of channel pruning methods calibrated with 128 samples from WikiText-2.

| Method | Sparsity | WinoG | HellaS | PIQA | ARC-e | ARC-c | Avg. |
|---|---|---|---|---|---|---|---|
| LLaMA-2 7B (original) | 0% | 69.06 | 75.99 | 79.11 | 74.58 | 46.25 | 69.00 |
| SliceGPT | 20% | 62.74 | 49.78 | 64.25 | 51.47 | 31.06 | 51.86 |
| ModeGPT | 20% | 68.03 | 69.05 | 74.05 | 69.07 | 42.06 | 64.46 |
| Our MLP pruning | 20% | 66.06 | 66.54 | 73.23 | 65.19 | 38.91 | 61.99 |
| Our MHA pruning | 21.02% | 66.93 | 69.64 | 73.94 | 63.97 | 42.24 | 63.34 |
| Our MHA + MLP Pruning | 21.07% | 68.03 | 71.64 | 76.17 | 68.98 | 44.28 | **65.82** |
| LLaMA-2 13B (original) | 0% | 72.22 | 79.39 | 80.47 | 77.48 | 49.23 | 71.76 |
| SliceGPT | 20% | 67.17 | 53.58 | 65.83 | 55.81 | 35.84 | 55.65 |
| ModeGPT | 20% | 70.32 | 68.96 | 74.53 | 74.07 | 46.16 | 66.81 |
| Our MHA + MLP Pruning | 21.07% | 71.43 | 75.26 | 77.58 | 71.68 | 45.39 | **68.94** |

channel pruning approach has a clear advantage over previous pruning methods. By combining the MHA pruning and MLP pruning, our method achieves the best performance, surpassing the previous channel pruning method, including SliceGPT (Ashkboos et al., 2024) and ModeGPT (Lin et al., 2024).

We further make ablations on the effectiveness of Nyström approximation. As shown in Table 10, Nyström approximation outperforms the channel selection only method, demonstrating the effectiveness of adjusting the down projection.

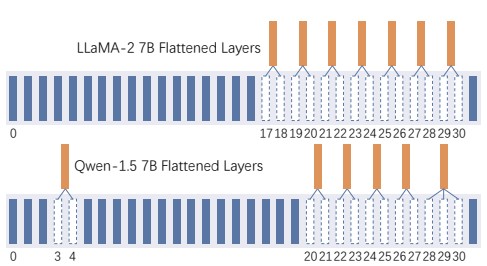 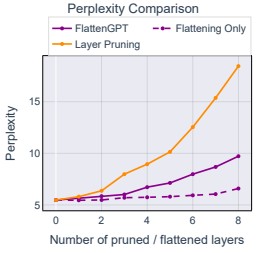 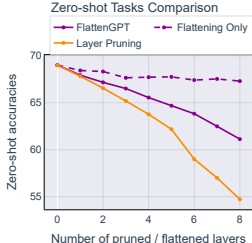

Figure 5: Flattened Layer indices.  Figure 6: Comparison of layer pruning and flattening.

Table 9: Zero-shot task performance of channel pruning methods calibrated with 128 samples from WikiText-2. † indicates fine-tuned on Alpaca (Taori et al., 2023) dataset. ModeGPT (Lin et al., 2024) employs Alpaca as the calibration dataset. LLM Surgeon (van der Ouderaa et al., 2024) does not show the results but claims that LoRA cannot improve the performance.

| Method | Sparsity | WinoG | HellaS | PIQA | ARC-e | ARC-c | Avg. |
|---|---|---|---|---|---|---|---|
| LLaMA-2 7B (original) | 0% | 69.06 | 75.99 | 79.11 | 74.58 | 46.25 | 69.00 |
| LLM-Pruner | 18.82% | 61.17 | 66.13 | 76.66 | 64.86 | 37.88 | 61.34 |
| LLM-Pruner† | 18.82% | 61.88 | 67.13 | 77.48 | 65.78 | 38.48 | 62.15 |
| LLM Surgeon | 20% | 66.30 | 71.30 | 77.09 | 71.36 | 41.89 | 65.59 |
| ModeGPT | 20% | 68.19 | 69.59 | 76.22 | 71.71 | 41.89 | 65.52 |
| ModeGPT† | 20% | 66.30 | 68.07 | 77.20 | 70.45 | 42.92 | 64.99 |
| FlattenGPT | 20% | 67.40 | 70.74 | 74.59 | 64.44 | 41.98 | 63.83 |
| FlattenGPT† | 20% | 68.75 | 73.01 | 74.97 | 67.40 | 45.05 | **66.24** |

## D.5 ADVANTAGES OF DEPTH COMPRESSION OVER WIDTH COMPRESSION

In this paper, we focus on the depth compression tasks. Although previous depth compression methods perform much worse than the width compression ones, FlattenGPT has built a novel approach to improve this performance greatly. In the main paper, we have shown that FlattenGPT achieves a better trade-off between performance and speed. In this part, we will further show that FlattenGPT shows promising performance compared with the latest width compression method after recovery fine-tuning. Table 9 shows the performance with or without RFT. LLM-pruner (Ma et al., 2023) shows little improvement with RFT. LLM Surgeon (van der Ouderaa et al., 2024) does not show the results, but it claimed that LoRA improves compression performance in the smallest OPT-125m model, but not in larger models. ModeGPT (Lin et al., 2024) even demonstrates performance loss after RFT, which illustrates that the model probably suffers from overfitting. FlattenGPT unifies the two tasks of deep compression and channel compression, making the pruned model more suitable for fine-tuning. This is more practical than previous pruning methods.

## D.6 LOCATIONS OF FLATTENED LAYERS

We show which transformer blocks are chosen to be flattened in Table 11. The location of flattened transformer blocks is highly consistent across various target models. The late blocks are almost flattened, except the last one or two, whereas the early blocks are rarely selected. This is related to the similarity distribution in the model, where the late blocks have more similar input features.

## D.7 DEPENDENCY ON CALIBRATION DATASET

We evaluate the dependency on the calibration dataset in Table 12. We use the calibration set size of 128 and sequence length of 2048 for WikiText-2 (Merity et al., 2016) and Alpaca datasets. The results show that WikiText-2 has a slightly better performance, probably due to the dataset quality. The alpaca dataset is not as representative as a high-quality dataset, thus the performance is slighter lower than WikiText-2.

Table 10: Comparison of channel selection and Nyström approximation.

| Method | Sparsity | WinoG | HellaS | PIQA | ARC-e | ARC-c | Avg. |
|---|---|---|---|---|---|---|---|
| Channel Selection | 20% | 66.46 | 65.48 | 71.22 | 63.13 | 39.93 | 61.24 |
| + Nyström approximation | 20% | 66.54 | 68.45 | 72.74 | 63.43 | 41.30 | 62.49 |

Table 11: Locations of flattened Transformer blocks with target sparsity of 20%.

| Models | Merged Layer Index |
|---|---|
| LLaMA-2 7B | [[17, 18], [19, 20], [21, 22], [23, 24], [25, 26], [27, 28], [29, 30]] |
| LLaMA-2 13B | [[23, 24], [25, 26], [27, 28], [29, 30], [31, 32], [33, 34], [35, 36], [37, 38]] |
| LLaMA-2 70B | [[14, 15], [46, 47], [49, 50], [51, 52], [54, 55], [57, 58], [59, 60, 61], [62, 63, 64], [65, 66, 67], [68, 69], [70, 71], [72, 73], [74, 75]] |
| LLaMA-3 8B | [[16, 17], [18, 19], [20, 21], [23, 24], [25, 26], [27, 28], [29, 30]] |
| Qwen-1.5 7B | [[3, 4], [20, 21], [22, 23], [24, 25], [26, 27], [28, 29, 30]] |
| Qwen-1.5 14B | [[7, 8], [10, 11], [19, 20], [24, 25], [26, 27], [28, 29], [30, 31], [32, 33], [34, 35], [36, 37]] |

## D.8 DEPENDENCY ON THE CALIBRATION DATASET SIZE

We test the size of the calibration dataset from 64 to 1024 samples as shown in Table 13. Results confirm that 128 samples suffice, as larger sets yield marginal gains ($< 0.2\%$).

## D.9 GENERALIZATION ON OTHER TASKS

We conduct experiments on InternVL-C 6B, which is a large vision transformer that exhibits a similar cross-layer similarity pattern to the LLMs. The results in Table 14 show that our method has good generalization ability on vision transformers. The multimodal transformers are usually composed of an LLM transformer and a vision encoder transformer. Therefore, it is reasonable to apply our method to the LLM and the vision encoder individually.

## D.10 GENERALIZATION BEYOND TRANSFORMER ARCHITECTURE

Considering the various architectures available, it is far beyond the scope of this paper. Yet we can provide an analysis of the generalization of our method. Since most architectures use skip connections, the flattening stage is very general and should work on these architectures as well. However, there are not always appropriate channel pruning methods for these architectures. If there is an appropriate channel pruning method, our method would work on various architectures. Besides, transformer is a widespread baseline for many tasks, and our experiments on multiple transformer architectures and tasks have shown the effectiveness of our method.

## E   LIMITATION

FlattenGPT provides a novel approach for fine-grained LLMs depth compression, yet there are still some limitations. First, FlattenGPT is performed on uniform architectures, where flattening will not change the model architecture significantly. It is not trivial to compress the hybrid architectures, such as a combination of transformer Vaswani et al. (2017) and mamba (Gu & Dao, 2023). However, it is still worth researching the fine-grained depth compression method, as layer pruning methods operate on a very high granularity and cause performance degradation. Second, we use one of the channel pruning methods to implement our FlattenGPT, while our framework is not constrained to specific channel pruning methods. Developing better channel pruning methods will improve our depth compression method as well.

Table 12: Results on different calibration dataset.

| Method | Dataset | PPL | Sparsity | WinoG | HellaS | PIQA | ARC-e | ARC-c | Avg. |
|---|---|---|---|---|---|---|---|---|---|
| LLaMA-2 7B (original) | - | 0% | 5.47 | 69.06 | 75.99 | 79.11 | 74.58 | 46.25 | 69.00 |
| FlattenGPT | WikiText-2 | 21.02% | 8.68 | 67.40 | 70.74 | 74.59 | 64.44 | 41.98 | 63.83 |
| FlattenGPT | Alpaca | 21.02% | 11.84 | 67.64 | 67.92 | 72.31 | 62.54 | 39.25 | 61.93 |

Table 13: The zero-shot accuracies on LLaMA-2 7B with different calibration dataset size.

| Num of Samples | 64 | 128 | 256 | 512 | 1024 |
|---|---|---|---|---|---|
| Accuracy | 61.17 | 62.49 | 62.25 | 62.58 | 62.60 |

Table 14: The zero-shot accuracies on InternVL-C 6B.

| Model | Method | IN-1K | IN-A | IN-R |
|---|---|---|---|---|
| InternVL-C | Dense | 83.2 | 83.8 | 95.5 |
| | ShortGPT | 79.7 | 57.9 | 90.4 |
| | FlattenGPT | 81.6 | 74.6 | 93.7 |

