# OpenReview forum: "FlattenGPT: Depth Compression for Transformer with Layer Flattening"
_ICLR.cc/2026/Conference — ICLR 2026 Conference Withdrawn Submission_

### Official Review · Reviewer_ZFYy · 2025-10-28

**Soundness:** 2
**Presentation:** 3
**Contribution:** 2
**Rating:** 4
**Confidence:** 3

**Summary:**

This paper introduces FlattenGPT, a new approach for depth compression of Transformer-based large language models. Instead of removing entire layers or pruning channels independently, the authors propose a two-stage framework that first flattens adjacent layers based on cross-layer similarity and then performs channel pruning to reduce parameter count and computation. The layer-flattening stage merges similar consecutive Transformer blocks by concatenating their attention and MLP parameters, effectively reducing network depth while keeping cross-layer knowledge. The subsequent pruning step ensures that the merged layers remain compact and computationally efficient.
The authors provide both theoretical analyses and extensive experiments on several LLMs. Results show that FlattenGPT achieves better accuracy-speed trade-offs than prior compression methods such as SLEB, ShortGPT, and BlockPruner, maintaining up to 96% of the original model performance with 1.3× inference speedup.

**Strengths:**

1.The concept of layer flattening—merging similar adjacent layers instead of deleting them—is both creative and intuitive. It fills an unexplored middle ground between existing pruning approaches.

2.The analysis of layer redundancy through variance and gradient norms provides strong motivation.

3.The experiments are broad, covering multiple LLM backbones, with consistent improvements in both speed and accuracy.

**Weaknesses:**

1. "However, these methods usually assign different pruning ratio for each layer". However, this is not accurate. Methods like Wanda and SparseGPT actually apply the same sparsity ratio across all layers.
2. The authors argue that FlattenGPT can "preserve the knowledge learned in all blocks", but in theory, channel pruning can also achieve this goal, since it retains important connections within each layer.
3. "which is caused by the residual path spanning the entire LLM. This similarity is particularly evident in LLMs, indicating that there is a certain amount of redundancy within them. ". However, this reasoning is questionable: even if layers exhibit similar features, that does not necessarily imply redundancy, since features are generated through complex cross channel interactions.
4. The method lacks novelty. It essentially performs layer merging before channel pruning, without introducing a fundamentally new design or principle.
5. Both Wanda and SparseGPT also support channel pruning, yet the paper does not provide a direct comparison under the same sparsity ratio, which would be necessary for a fair evaluation.
6. The paper lacks ablation studies. Specifically, it should include an experiment that removes the Iterative Layer Flattening stage and keeps only Channel Pruning to isolate and verify the real contribution of the flattening step.

**Questions:**

1. The authors argue that channel pruning “cannot compress model depth,” suggesting this is a limitation. But why is not reducing depth necessarily a disadvantage? Channel pruning may still provide efficient compression without altering model depth.
2. How sensitive is the flattening process to the choice of similarity metric? Would cosine similarity always be the best option?
3. Is the proposed framework compatible with structured sparsity training or quantization techniques?
4. The paper claims that block pruning may remove useful knowledge and cause performance degradation. But depth pruning also removes parameters—why would it not suffer from the same issue?

---

### Official Review · Reviewer_XPer · 2025-10-31

**Soundness:** 3
**Presentation:** 2
**Contribution:** 3
**Rating:** 4
**Confidence:** 4

**Summary:**

FlattenGPT is a method that achieves the same effect as Transformer layer pruning, but concatenates non-adjacent groups of 2 similar adjacent layers in order to jointly prune them to achieve a single layer. The pruning relies on attention head importance-based pruning to reduce 2n heads to n, and Nystrom approximate based channel pruning for MLP layers. Results show accelerated inference compared to width-pruning strategies, and improved performance compared to other depth pruning strategies across 7B-70B range LLMs.

**Strengths:**

1. The results comparing this work to prior methods like SLEB, LaCO, BlockPruner, and ShortGPT show improved performance on zero-shot tasks and perplexity, while maintaining solid latency and throughput results.
2. The empirical comparision of the residual norm vs the MHA/MLP norm to motivate the paper is a nice contribution demonstrating a potential cause of layer redundancy.
3. The proposed method is straightforward, relatively simple, and easy to understand and implement. The basic method of concatenating layers and then trimming to a single layer width can be extended beyond the proposed channel pruning methods stated in this work.

**Weaknesses:**

1. Part of the motivation of this work discusses channel pruning as being suboptimal because it may result in differently sized transformer layers, whereas FlattenGPT preserves the Transformer layer structure, but with fewer layers. It is not clear In what specific sense is reducing the width/size of weights strictly worse than removing full layers. While speed vs channel pruning is stated, I am not understanding the hyperparameter tuning and/or model deployment issues with channel pruning.
2. A major contribution of the work is this depth analysis in section 2, but some of the key assumptions, like distribution of activations, are not well motivated. Why are normal distributions an okay assumption to make here to model the redundancy across depth? In Figure 2, the norms of MHA/MLP grow with the layer index; is this captured in the model?
3. The related work section address depth pruning, but does not seem to adequately address prior layer merging papers beyond just some citations, and this section appears a bit unfinished as well (line 464 cutoff). More discussion about these very related works would be helpful to contextualize the novelty and contribution of this work.

**Questions:**

1. What is the effect of flattening without pruning? An interesting and helpful ablation that could strengthen the work would measure how much performance is lost during flattening, and how much more performance is lost during channel pruning.
2. Why does it follow from Theorem of 2.2 that for large L, the gradient of the deeper layers is dominated by the identity mapping? Even if the variances grow between linearly and exponentially, why don’t we have to worry about their raw value being less than 1?
3. Is the layer selection algorithm designed to only allow pairs of non consecutive layers to be merged?
4. What exactly is the issue with LoRA hyperparameters if the channel width is different per layer? Can you provide an example or intuition?

Typos:

Similicity -> simplicity line 130

Detailes -> details line 373

---

### Official Review · Reviewer_2TKX · 2025-10-31

**Soundness:** 3
**Presentation:** 1
**Contribution:** 3
**Rating:** 4
**Confidence:** 3

**Summary:**

This manuscript presents a procedure for (sequentially) flattening a transformer architecture and then pruning the resulting layers in a structured manner. The first step is achieved by considering similarities between layers and the second by a Nystrom style scheme. Numerical results are provided that illustrate the efficacy of the scheme.

**Strengths:**

The method provided in the manuscript seems sensible. The strength/innovation is mostly in the high level algorithm and layer merging strategies; the structured pruning strategy seems less novel/interesting (as the manuscript notes, other strategies could be used).

In particular, the results seem good when compared with alternative depth compression strategies and competitive with pure structured pruning strategies (though in some places this comparison is a bit trickier particularly if more expensive schemes are allowed).

**Weaknesses:**

I think that the main weakness of the manuscript in its current form is the presentation. This manifests in two forms: (1) a significant number of grammatical errors that need to be addressed and (2) a lack of precision in various places that could lead to confusion and/or makes it hard to interpret results.

The first point is important to address but it's also clear what is needed; the second point is more significant. To illustrate:

I do not really think Theorems 2.1 and 2.2 add/say much. In addition they are imprecisely stated. For example, how is it reasonable to assume $\tilde{H}$ is normally distributed when its distribution is a consequence of the input distribution and layer structure (which may not even allow for such a distribution). I really think that the manuscript would be better served by bases its claims on (perhaps expanded) empirical evaluation rather than theorems that are clearly not applicable (or perhaps even sensible/illustrative).

Similarly, the Nystrom scheme is somewhat imprecisely outlined (see a question below). For example, line 2 of the algorithm is not really stated clearly: a matrix is assigned to the set of leverage scores? (I know what is intended, but only because I know what leverage scores are a priori.) I also don't see the value of Theorem 3.1, it's just a fact about least squares (as noted in the appendix) with the present manuscripts notation. Framed as a Theorem (and the main text could even be read in a manner that implicitly suggests it is new) is odd. It also takes up space that, e.g., could be used to better describe the method.

While it is reasonable to try pruning to the original width (per section 3.3) this seems quite limiting. Yes, there may be choices in the original architecture that are tuned to hardware performance, but typically such choices are not unique and there is definitely a bit of a design space there that seems worth exploring. At present the method would either "work satisfactorily" or result in and "unacceptable model;" there is no way to trade off computational efficiency for accuracy/performance.

Some minor notes:

- Figure 2(a) needs to be more precise in exactly what it is plotting.

- In fact, that statement is true for all of Figure 2; for example in Fig. 2 (d) its not clear which method is being plotted where and what precisely is the "experiment" (for example, two papers are cited in the discussion about the figure in the main text and the caption lacks detail).

- It would seem natural to validate the flattening phase (before any pruning) with some experiments (i.e., how much is lost at that step); this would help contextualize the pruning step in terms of "what is possible" (maybe I missed these in the appendix).

**Questions:**

- Why the Nystrom style scheme? Other closely related schemes that use "column selection" exist (see, e.g., [Jerry Chee, Megan Flynn (née Renz), Anil Damle, and Christopher De Sa. 2022. Model preserving compression for neural networks. In Proceedings of the 36th International Conference on Neural Information Processing Systems 2022]). It is not clear a "new" scheme is necessary here. In some ways this is not so important if the performance is acceptable, but it does have side effect that, as presented, I think it would actually be hard to implement the scheme outlined due to a lack of precision in the description.

- Why is perplexity not reported in some of the results (e.g., later in appendix when comparing with LLM-surgeon)?

- I don't get the "bold" strategy for the plots; for example in Table 4 there are places where sliceGPT has lower perplexity but the FlattenGPT number is in bold. Moreover, bold text is used very sporadically, what is it meant to indicate?

---

### Official Review · Reviewer_FAYR · 2025-11-01

**Soundness:** 2
**Presentation:** 3
**Contribution:** 2
**Rating:** 4
**Confidence:** 3

**Summary:**

This paper introduces FlattenGPT, a new depth compression technique for transformers that merges adjacent layers (layer flattening) and then applies channel/head pruning to maintain the original architectural shape. The method aims to combine the benefits of depth pruning (reduced latency) and channel pruning (preserved performance). FlattenGPT select merge candidates using cross-layer feature similarity, then performs pruning guided by Nystrom approximation to retain fine-grained information. Experiments on several large-scale LLMs (LLaMA, Qwen and mores) show better trade-offs between accuracy and inference speed than existing baselines such as SLEB, ShortGPT, and BlockPruner.

**Strengths:**

(1) Maintains original block width and head count, ensuring easy deployment and LoRA compatibility.

(2) Demonstrates consistent gains in throughput and latency without severe accuracy drops.

(3) Builds on empirical findings on strong cross-layer redundancy in transformer residual paths.

**Weaknesses:**

(1) Relies on small calibration sets and greedy similarity-based layer pairing, which may be unstable.

(2) Evaluation focus on decoder-only LLMs, applicability to encoder or encoder-decoder architectures (like VLMs) is unverified.

(3) Reported accretion results may  vary across GPU architectures or models or inference backends.

(4) Analytical justification of flattening equivalence is mostly empirical and heuristic.

**Questions:**

Please refer to the weaknesses part.

---

### Note · Authors · 2026-01-09

I have read and agree with the venue's withdrawal policy on behalf of myself and my co-authors.